# The contribution of genetic determinants of blood gene expression and splicing to molecular phenotypes and health outcomes

Alex Tokolyi [1,21], Elodie Persyn [2,3,4,21], Artika P. Nath[4,5], Katie L. Burnham [1], Jonathan Marten[2], Thomas Vanderstichele[1], Manuel Tardaguila [1,6], David Stacey [2,7,8], Ben Farr[1], Vivek Iyer[1], Xilin Jiang [2,3,9], Samuel A. Lambert[2,3,4,10], Guillaume Noell[1], Michael A. Quail [1], Diana Rajan[1], Scott C. Ritchie [2,3,4,5,10,11], Benjamin B. Sun [12], Scott A. J. Thurston [1], Yu Xu [2,3,4], Christopher D. Whelan [13], Heiko Runz[12], Slavé Petrovski [14,15], Daniel J. Gaffney[1,16], David J. Roberts[17,18], Emanuele Di Angelantonio [2,3,6,10,11,19], James E. Peters [20], Nicole Soranzo[1,6], John Danesh[1,2,3,10,11,19], Adam S. Butterworth [2,3,10,11,19], Michael Inouye [2,3,4,5,10,11,21], Emma E. Davenport[1,21] & Dirk S. Paul [2,3,14,21] ✉

The biological mechanisms through which most nonprotein-coding genetic variants affect disease risk are unknown. To investigate gene-regulatory mechanisms, we mapped blood gene expression and splicing quantitative trait loci (QTLs) through bulk RNA sequencing in 4,732 participants and integrated protein, metabolite and lipid data from the same individuals. We identified *cis*-QTLs for the expression of 17,233 genes and 29,514 splicing events (in 6,853 genes). Colocalization analyses revealed 3,430 proteomic and metabolomic traits with a shared association signal with either gene expression or splicing. We quantified the relative contribution of the genetic effects at loci with shared etiology, observing 222 molecular phenotypes significantly mediated by gene expression or splicing. We uncovered gene-regulatory mechanisms at disease loci with therapeutic implications, such as *WARS1* in hypertension, *IL7R* in dermatitis and *IFNAR2* in COVID-19. Our study provides an open-access resource on the shared genetic etiology across transcriptional phenotypes, molecular traits and health outcomes in humans (https://IntervalRNA.org.uk).

The majority of genetic variants associated with common diseases and other complex traits identified through genome-wide association studies (GWAS) lie in nonprotein-coding sequences[1]. Consequently, the molecular mechanisms that underpin many of these genotype–phenotype associations are unclear. Molecular quantitative trait locus (QTL) mapping studies, which identify genetic determinants of transcript, protein or metabolite abundance, can address this knowledge gap by identifying the molecular intermediaries that mediate genetically driven disease risk. These studies can provide specific hypotheses for functional validation experiments[2,3].

Molecular QTL data can be used for a range of biomedical applications. For example, they have the potential to identify and validate new therapeutic targets and pathways, inform about the biological mechanisms of drug action and safety, highlight new therapeutic indications and reveal clinically relevant biomarkers[4–6].

Many previous studies have carried out QTL mapping within a single molecular domain such as gene or protein expression[7–12]. However, QTL data from multi-omic modalities are needed to fully understand the causal molecular chain of events from genetic variation to complex trait phenotypes[13]. Moreover, the availability of multimodal data in a

---

single population sample enhances the interpretation and validity of causal inference analyses. For example, mediation analysis in a single cohort presents a strategy for identifying phenotypes that share a common genetic pathway and quantifying the proportion of the total genetic effect on those phenotypes[14].

Here we use the INTERVAL study[15,16], a bioresource of approximately 50,000 blood donors with extensive multidimensional 'omics' profiling, to identify gene expression and splicing QTLs based on peripheral blood RNA sequencing (RNA-seq; $n$ = 4,732 individuals). Then, we integrate the QTL data with additional molecular QTL data derived from the same study (Fig. 1). These data include plasma protein levels measured through an antibody-based proximity extension assay (Olink Target panels, $n$ = 4,662–4,981 individuals)[17,18] and an aptamer-based multiplex protein assay (SomaScan v3, $n$ = 3,301)[5], as well as serum metabolite levels measured using an untargeted mass spectrometry platform (Metabolon HD4, $n$ = 14,296)[10] and a nuclear magnetic resonance spectroscopy platform (Nightingale Health, $n$ = 40,849)[19,20].

Our data reveal genetic effects on the expression and splicing of local and distant genes. We assess shared genetic etiology across molecular traits and health outcomes using statistical colocalization. We further investigate the genetic effects on downstream molecular phenotypes through transcriptional events by conducting mediation analyses. Based on these analyses, we develop an open-access portal that enables exploration of this compendium of molecular QTLs (https://IntervalRNA.org.uk).

## Results

### Genetic regulation of local gene expression and splicing

We performed bulk RNA-seq on peripheral blood collected from 4,732 blood donors recruited as part of the INTERVAL study (Methods). The expression levels of 19,173 autosomal genes and 111,937 de novo transcript splicing phenotypes (herein referred to as 'splicing events') from differential intron usage ratios in 11,016 genes were quantified. Then, we mapped local (cis) expression QTLs (eQTLs) within ±1 Mb of the transcription start site (TSS) and splicing QTLs (sQTLs) within ±500 kilobase pairs (kb) of the center of the spliced region.

We identified 17,233 genes (89.9% of the 19,166 tested) with at least one significant cis-eQTL (cis-eGene) at a false-discovery rate (FDR) < 0.05 (Supplementary Tables 1 and 2; Methods). Stepwise conditional analyses for each cis-eQTL revealed 56,959 independent signals (53,457 unique lead variants), with a median of three independent signals per gene (range = 1–23; Supplementary Tables 1 and 3; Methods). We compared our results to those from the eQTLGen consortium study given in ref. 9 ($n$ = 31,684 individuals). $z$ scores from eQTL lead SNPs were highly correlated between these studies (Pearson's $r$ = 0.9; Supplementary Fig. 1 and Supplementary Tables 2 and 4), highlighting the consistency of eQTL discovery results across independent datasets and mapping technologies.

Next, we identified 29,514 splicing events with a cis-sQTL at FDR < 0.05 (Supplementary Tables 1 and 5). These splicing events with a cis-sQTL were mapped to 6,853 genes (cis-sGenes) with a median of three splicing events observed per cis-sGene (range = 1–128). This included 543 cis-sGenes that were not identified as cis-eGenes. Across all splicing events with cis-sQTLs, these had a median length of 1,549 base pairs (bp) and excised a protein-coding sequence in 32.4% of cases (the remainder related to intronic and UTR excisions). The median distance from the cis-sQTL lead variants to the center of the splicing event was 187 bp upstream, with lead variants forming a bimodal distribution around the start and end of the sGene (Fig. 2a).

After conditional analysis for each cis-sQTL, we identified 47,050 independent signals (34,205 unique lead variants), with a median of one independent signal per cis-sQTL (range = 1–20; Supplementary Tables 1 and 6). To characterize independent variant effects on transcript splicing, we compared primary (that is, the most significant

independent QTL) and secondary (that is, all other independent QTLs) cis-sQTLs. Primary cis-sQTL signals were enriched within the gene body of sGenes compared to secondary signals ($P$ = 2.84 × 10$^{-314}$, chi-squared test; Fig. 2a and Supplementary Fig. 2). Primary cis-sQTL signals were more enriched toward the transcription end site (median of 17.36 kb downstream of the TSS) compared to cis-eQTLs with a median of 5.51 kb downstream of the TSS ($P$ = 8.42 × 10$^{-259}$, two-sided Wilcoxon test; Supplementary Fig. 2). These observations align with previous analyses for isoform ratio QTLs[21]. Next, we compared the identified sGenes with those from the Genotype-Tissue Expression (GTEx) Consortium[22] whole-blood dataset ($n$ = 670 individuals). Of the 3,013 sGenes discovered by GTEx, 89.0% of the 2,677 we tested were also found as sGenes in our analysis, in addition to 4,470 new sGenes (Supplementary Table 7). These results demonstrate the value of quantifying de novo splicing excision events and the substantially larger sample size.

For a given gene, to test whether corresponding cis-eQTLs and cis-sQTLs were underpinned by the same genetic variant, we performed colocalization analyses. This revealed 3,979 genes (of 6,252 tested) with colocalized signals (Methods). We found that 49.0% ($n$ = 13,490) of tested splicing events had sQTLs that colocalized with an eQTL for the same gene (Supplementary Table 8). However, of the eQTL-colocalizing splicing events with multiple independent sQTL signals, 82% had additional sQTL loci that did not colocalize with eQTLs. Splicing events with sQTL that did not colocalize with an eQTL were located further downstream of the TSS (median 20.33 kb downstream) compared to sQTL signals that did colocalize (median 12.61 kb downstream; $P$ = 9.8 × 10$^{-70}$, two-sided Wilcoxon test; Supplementary Fig. 3).

### Genetic effects on distal gene expression and splicing

Next, we investigated the distal (trans) regulatory effect of genetic variants (>5 Mb from the TSS/splicing event). First, we performed an untargeted, all-versus-all trans-eQTL and trans-sQTL analysis. We found a high correlation of trans-eQTL $z$ scores for SNP–gene pairs also tested by the eQTLGen consortium study[9] (Pearson's $r$ = 0.9; Supplementary Fig. 4). As our study has lower statistical power than eQTLGen, we focused on replicating the 2,924 most significant trans-eQTL associations from eQTLGen with $P$ < 1 × 10$^{-20}$. Of these, we replicated 63% at $P$ < 1 × 10$^{-6}$ in INTERVAL. We note that the incomplete overlap may be due to differences in data analysis strategies, including accounting for blood cell counts in the association model[9].

Given the extreme multiple testing burden for genome-wide trans-QTL analyses, we focused on the 53,457 conditionally independent lead cis-expression SNPs (eSNPs), as these provide a potential mechanism through which a cis-acting variant can also affect genes in trans. In this targeted analysis, we identified 2,058 trans-eGenes at the Bonferroni-corrected threshold of $P$ < 5 × 10$^{-11}$ (Fig. 2b and Supplementary Tables 1, 2 and 9). These trans-eQTLs corresponded to 2,498 cis-eQTLs, with a median of three trans-eGenes per cis-eQTL (range = 1–284). Some of the cis-eGenes were associated with many trans-eGenes, such as PLAG1 ($n$ = 284 genes), HYMAI ($n$ = 284) and FUCA2 ($n$ = 267). Cis-eGenes with a concurrent trans-association were significantly enriched for 32 gene ontology (GO) terms, compared to all cis-eGenes. Most of the terms related to transcription regulation and immune response, with 'metal ion binding' showing the strongest enrichment ($P$ = 2.6 × 10$^{-30}$; Supplementary Table 10). To further explore these transcriptional regulation mechanisms, we annotated the genes using the Human Transcription Factors database[23]. We found a significant enrichment in sequence-specific transcription factors, representing 14.3% of all cis-eGenes with a trans-association (357/2,498, $P$ = 1.83 × 10$^{-38}$; Methods). We investigated protein domain annotations for the observed transcription factors and detected a significant enrichment for the C2H2 zinc finger domain ($P$ = 9.74 × 10$^{-9}$ after Bonferroni multiple-testing correction), specifically with the Krüppel-associated box domain ($P$ = 3.04 × 10$^{-10}$; Supplementary Fig. 5). For example, the PLAG1 gene, which is an important regulator of the

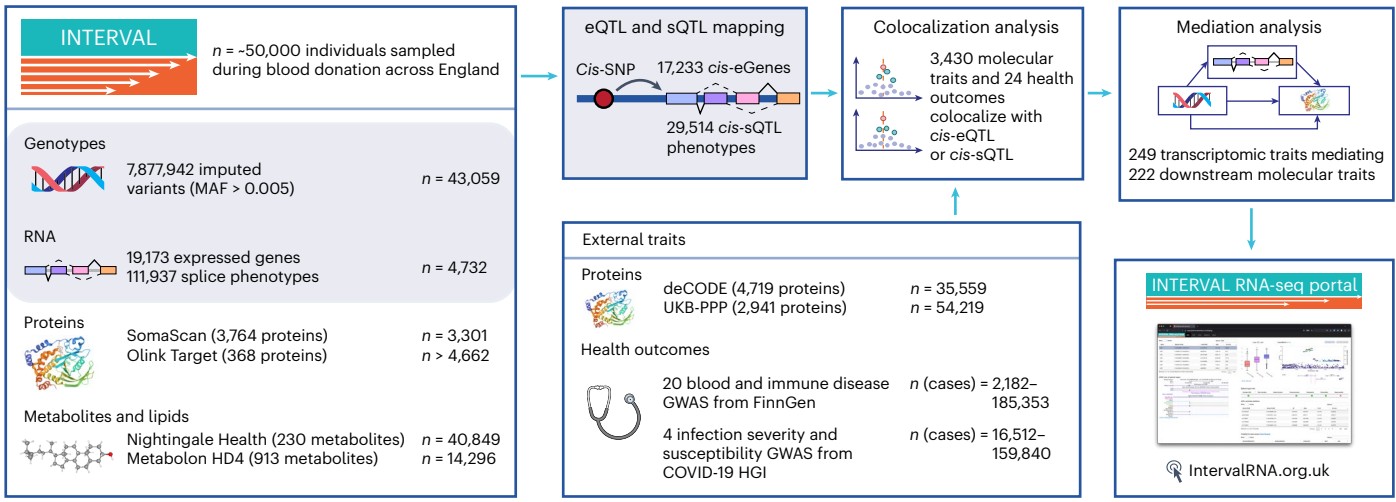

**Fig. 1 | Overview of the multi-omic data available in the INTERVAL study and external cohorts, as well as the main analytical approaches.** The images depicting 'genotypes' (the figure is created with NIAID NIH Bioart) and 'proteins' (Protein Data Bank (PDB) code 2F6W) were reproduced from public databases. COVID-19 HGI, COVID-19 host genetics initiative; MAF, minor allele frequency.

human hematopoietic stem cell dormancy and self-renewal[24], codes for a protein with a C2H2 zinc finger domain. We also noted the same enrichment for *trans*-eGenes ($P = 5.73 \times 10^{-5}$), although the molecular mechanisms are unclear and need further investigation.

To uncover genetic expression effects impacting distal downstream transcript splicing, we performed a targeted *trans*-analysis using the same 53,457 conditionally independent lead *cis*-eSNPs as in the *trans*-eQTL analysis. The analysis identified significant *trans*-associations for 644 splicing events (209 *trans*-sGenes) at the Bonferroni-corrected threshold of $P < 8.36 \times 10^{-12}$. This comprised 758 unique *trans*-splicing SNPs (sSNPs), corresponding to 566 *cis*-eGenes (Fig. 2b and Supplementary Tables 1 and 11). Of the 644 splicing events regulated in *trans*, 240 (in 91 genes) were not observed to be regulated in *cis*, increasing the total number of splicing events with QTLs. We observed 11 *cis*-eGenes that were implicated by their *cis*-eQTLs in the regulation of more than ten sGenes in *trans*. For example, we observed that the *cis*-eQTL for the RNA-binding splice factor *QKI* was associated with 18 sGenes in *trans* (the most of any eGene; Fig. 2c). Across all tissues in GTEx, there were only 29 *trans*-sQTL associations, of which only two were present in whole blood, that is, the *trans*-splicing of *FYB1* via the *QKI cis*-eQTL and the *trans*-splicing of *ABHD3* for which they did not detect an associated *cis*-effect for the *trans*-sSNP[22]. Here we replicated both of these previous *trans*-sGene observations. For *ABHD3*, we demonstrate in addition that this *trans*-sSNP is also a *cis*-eSNP for the splicing factor *TFIP11* and its antisense long noncoding RNA *TFIP11-DT*, potentially regulating the splicing of this gene in *trans*. Reports of the preliminary overlap of *trans*-sQTL associations with previous experimental validation are given in Supplementary Note and Supplementary Table 12. Across the whole dataset, *cis*-eGenes of *trans*-sSNPs were significantly enriched for ten GO terms, including 'nucleosome assembly' ($P = 2.78 \times 10^{-6}$) and 'RNA polymerase II activity' ($P = 1.40 \times 10^{-5}$; Supplementary Table 10).

### Shared genetic etiology across molecular traits

We next compared transcriptional QTLs to the other omics QTLs derived from subsets of participants from the INTERVAL study. These data include plasma protein QTLs (pQTLs) from the Olink Target and SomaScan panels, as well as metabolite QTLs (mQTLs) from the Metabolon and Nightingale Health platforms.

To determine whether genetic signals at a given locus across omics layers reflect shared genetic or distinct causal variants, we performed statistical colocalization analyses (Methods). These analyses revealed colocalization between either a *cis*-eQTL or *cis*-sQTL and *cis*-QTL for 120 Olink-measured proteins (65.9% of analyzed proteins), 404 SomaScan-measured proteins (63.7%), 224 Nightingale-measured metabolites (99.1%) and 495 Metabolon-measured metabolites (81.5%; Fig. 3a and Supplementary Tables 13–16). We found colocalized signals across all assessed proteomic and metabolomic traits for 1,229 *cis*-eGenes and 649 *cis*-sGenes (1,516 unique genes). For Olink- and SomaScan-measured proteins, genetic effect directions were more consistent ($P = 5.4 \times 10^{-10}$, one-sided Fisher's exact test) for colocalizing eQTL–pQTL pairs (78.9% with consistent effect directions) than noncolocalizing pairs (59.0%). The uncoupling of eQTLs and pQTLs has previously been observed[25] and could be due, for example, to post-transcriptional or post-translational mechanisms.

Of the 99 eQTL–pQTL pairs (364 sQTL–pQTL pairs) analyzed for colocalization in both Olink and SomaScan platforms, we found that 45 (127) had a colocalized signal in both platforms, 19 (57) on the Olink platform only and 9 (41) on the SomaScan platform only (Supplementary Tables 17 and 18). We annotated these colocalization results with cross-assay correlations reported previously[26] and found significantly higher cross-assay correlations for eQTL/sQTL–pQTL pairs with a colocalized signal for both platforms compared to eQTL–pQTL pairs with a colocalized signal for only one platform (eQTL–pQTL pairs, $P = 3.1 \times 10^{-4}$; sQTL–pQTL pairs, $P = 1.4 \times 10^{-7}$; one-sided Wilcoxon rank-sum test). This indicates that the differences we observed in colocalization results might be due to differences in protein measurements between the two platforms.

Next, we created a network to explore and visualize the interconnectedness among colocalized transcriptional and molecular phenotypes (Fig. 3b), linking each phenotype by their colocalizations. For example, we found seven splicing events in the *OAS1* gene with *cis*-sQTLs that colocalized with both the *cis*-eQTLs for this gene and the OAS1 pQTLs.

To investigate the potential mechanisms by which genetic variants impact protein levels through splicing, we annotated the protein domains affected by splicing events. We observed that nearly half of splicing events that colocalized with pQTLs (41.0%, 401 of 977) excised annotated protein-coding sequences. Splicing has been shown to modulate circulating protein levels through changes in secretion by the inclusion or exclusion of transmembrane domains[27]. This is exemplified by a splicing event that removes exon 6 of the *FAS* gene, a cell surface receptor for the FAS-ligand (FASL) cytokine. The resulting protein, lacking a transmembrane domain, is secreted[28] and competitively inhibits

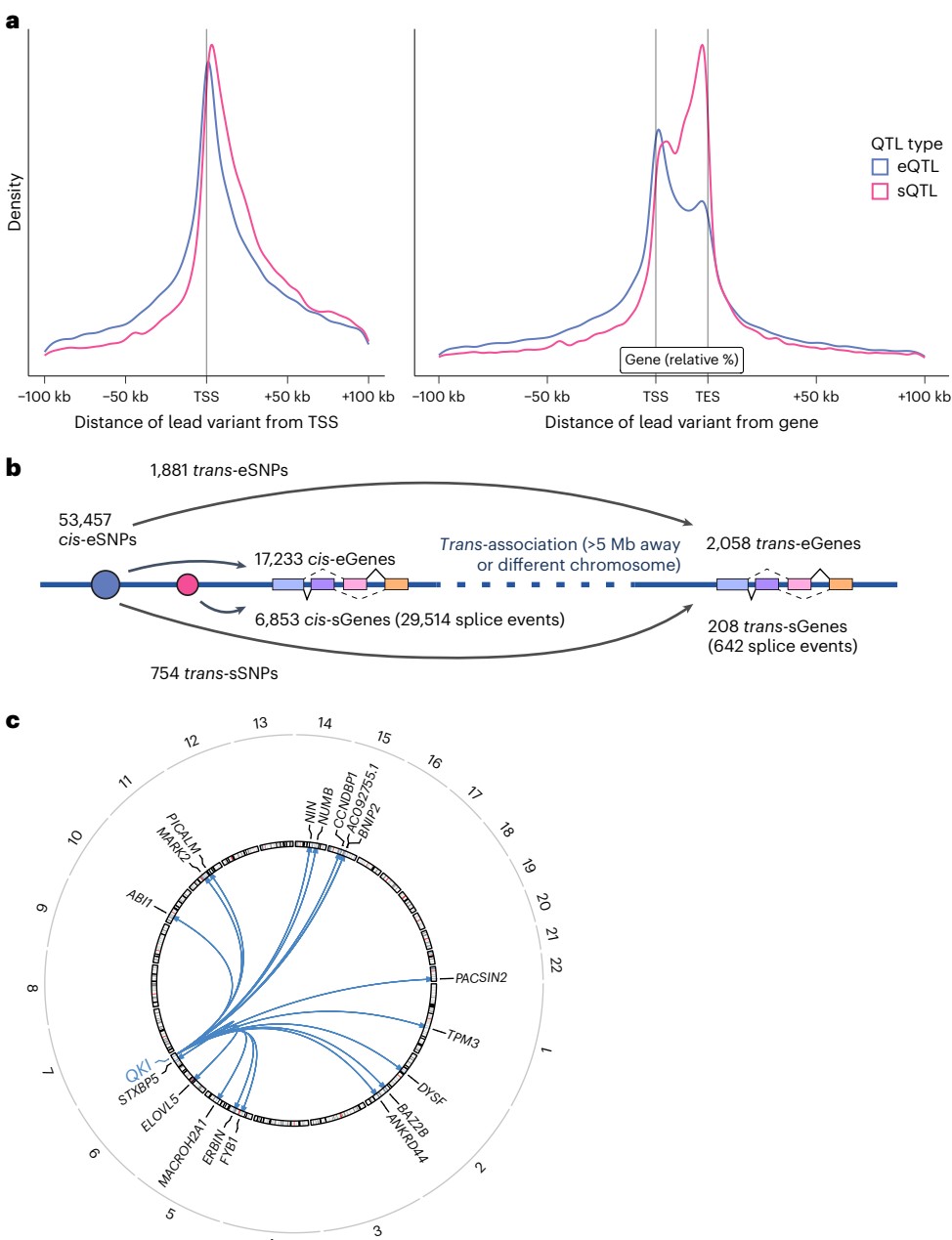

**Fig. 2 | Genetic influences on gene expression and splicing. a**, Distribution of lead variants at *cis*-eQTLs and *cis*-sQTLs around the TSS and gene body (normalized to the median gene length of 24 kb). **b**, Schematic representation of the *trans*-QTL mapping analysis approach and summary of the QTL discovery results. **c**, Circos plot of the *trans*-splicing of 18 sGenes by the *cis*-eQTL for *QKI*. TES, transcription end site.

FASL binding, leading to decreased apoptosis. We identified both *cis*-eQTLs for *FAS* and *cis*-sQTLs for this splicing event, but these signals were distinct and did not colocalize (maximum posterior probability = 0.02). The *cis*-sQTLs for excision of the transmembrane domain strongly colocalized with the pQTL (posterior probability = 1.00). Similarly, the interleukin-6 and interleukin-7 receptors (IL-6R and IL-7R, respectively) have previously been reported to produce secreted isoforms through the excision of transmembrane domains[29,30]. Here we show that the pQTLs for IL-6R and IL-7R colocalized with *cis*-sQTLs excising these transmembrane domain-encoding exons, in the absence of *cis*-eQTL colocalization (Fig. 3b). This observation emphasizes the role of transcript splicing as a mechanism independent of total transcript abundance through which genetic variation can modify downstream molecular phenotypes. Furthermore, we observed a pQTL colocalizing with an sQTL for the excision of a transmembrane domain in the

encoding messenger RNA (mRNA) in 69 proteins (98 unique splicing events), with 60.2% of these independent sQTL signals (n = 100/166) not colocalizing with eQTLs for the same gene (Supplementary Table 19). For example, this is observed in α-1 antitrypsin encoded by *SERPINA1* and apolipoprotein L1 encoded by *APOL1*. Of these 69 transmembrane proteins, the majority were annotated as being single-pass, with only four (ENTPD1, ADGRE2, ADGRE5 and ADGRE1) being multipass transmembrane proteins.

To maximize statistical power for colocalization, we extended our analyses to the SomaScan-pQTL and Olink-pQTL datasets from deCODE[8] (n = 35,559 individuals and n = 4,719 proteins) and the UK Biobank Pharma Proteomics Project[12] (UKB-PPP; n = 54,219 individuals and n = 2,941 proteins), respectively. Colocalization analyses were performed between 1,608 Olink- and 1,410 SomaScan-measured proteins and our transcriptional phenotypes, increasing the discovery of

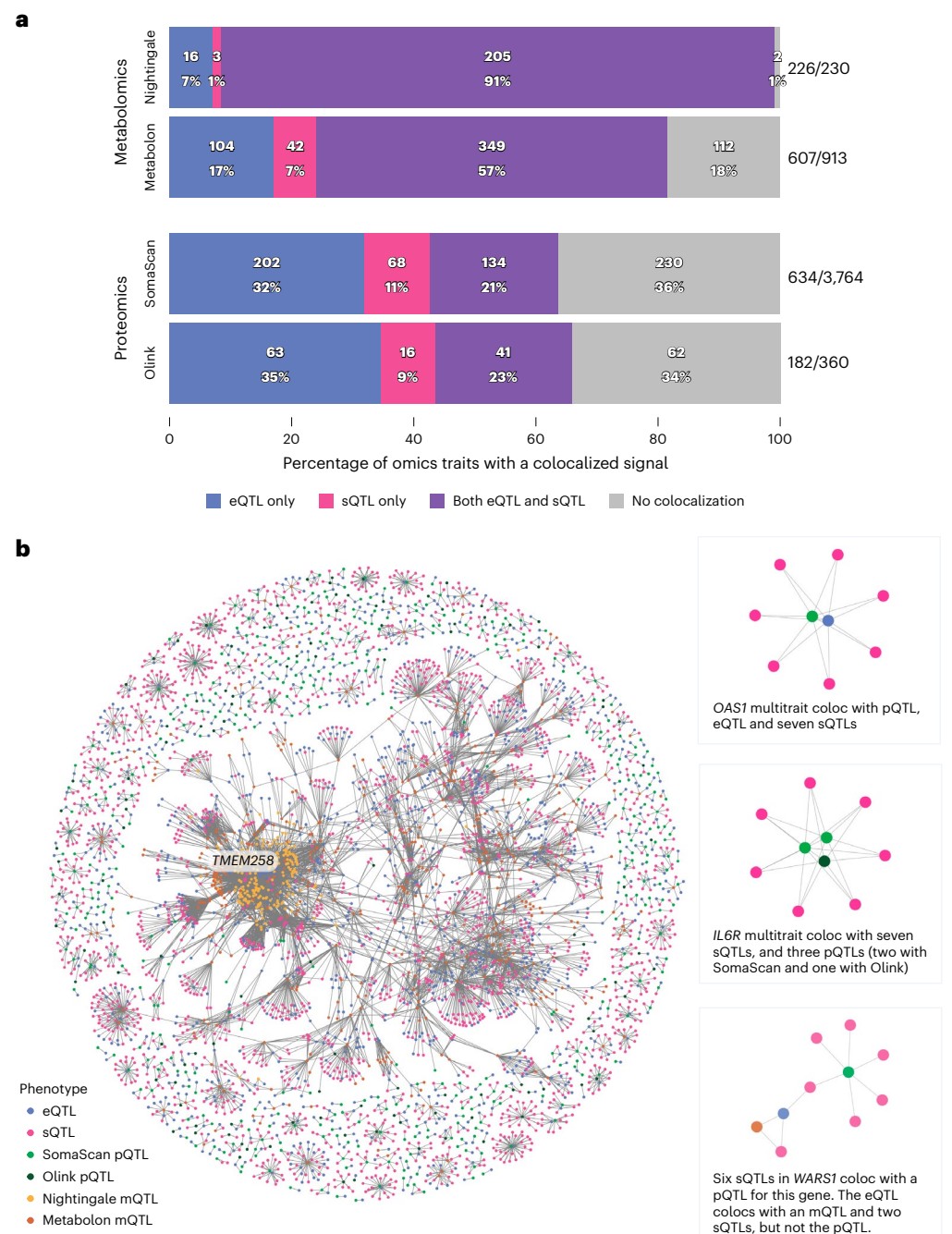

**a**

**Fig. 3 | Colocalization analyses of *cis*-eQTL and *cis*-sQTL with other molecular phenotypes. a**, Barplot of the percentage of omics traits with a colocalized association signal with a *cis*-eQTL or/and a *cis*-sQTL. **b**, Network graph of all pairwise colocalization results. Highlighted examples on the right-hand side include *OAS1*, *IL6R* and *WARS1*.

pQTL–eQTL/sQTL colocalizations from 120 to 1,203 Olink-measured proteins and from 404 to 984 SomaScan-measured proteins. We observed a substantial overlap of eGenes and splicing events with QTLs colocalizing between our internal and the larger external pQTL cohorts. In UKB-PPP, we replicated 95.1% and 79.3% of eQTLs and sQTLs colocalizations, respectively, and in deCODE, 87.0% and 80.3% of eQTLs and sQTLs, respectively (Supplementary Tables 13–15; web portal).

**Mapping causal transcriptional events on molecular phenotypes**

To assess the causality of the transcriptional phenotypes on downstream molecular phenotypes, we performed mediation analyses

focusing on colocalizing molecular traits assayed in the INTERVAL study (Fig. 4a; Methods). The expression of 143 *cis*-eGenes significantly mediated the effect of 413 *cis*-eSNPs on 202 downstream molecular phenotypes, including 101 SomaScan-measured proteins, 54 Olink-measured proteins, 39 Nightingale-measured metabolites and 8 Metabolon-measured metabolites. In total, this comprised 525 significant eQTL mediation models (variant-gene-molecular phenotype triplets; Fig. 4b). Similarly, we observed 106 splicing event phenotypes in 47 sGenes that significantly mediated the effect of 152 *cis*-sSNPs on 50 downstream molecular phenotypes, including 32 SomaScan-measured proteins, 16 Olink-measured proteins, 1 Nightingale-measured metabolite and 1 Metabolon-measured

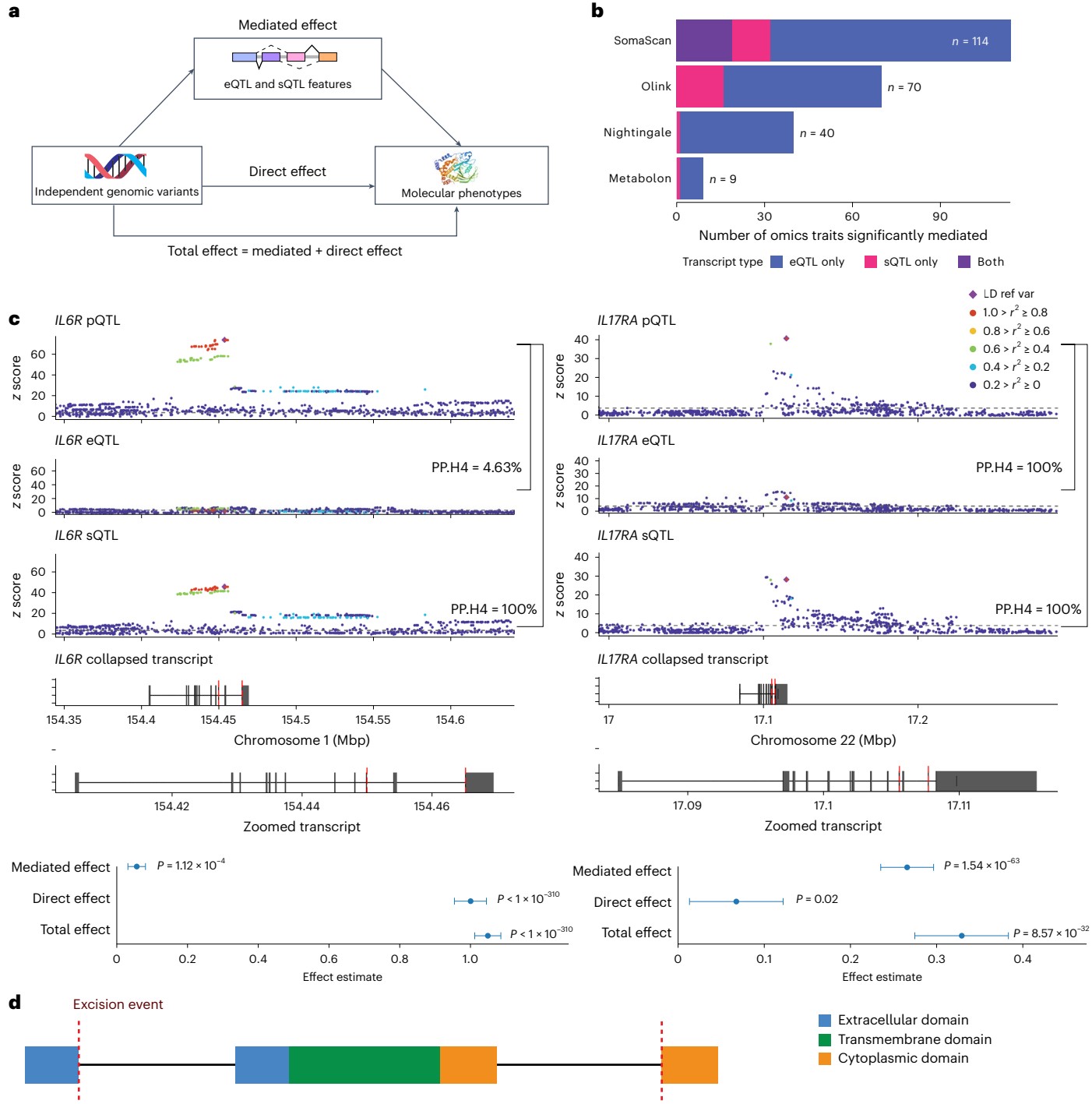

**Fig. 4 | Mediation analyses of molecular phenotypes with transcriptional QTLs. a**, Schematic representation of the tested mediation model, for which eQTL and sQTL phenotypes mediate the relationship between genomic variants and levels of molecular phenotypes. The images depicting 'independent genomic variants' (the figure is created with NIAID NIH Bioart) and 'molecular phenotypes' (PDB code 2F6W) were reproduced from public databases. **b**, Total number of detected molecular phenotypes mediated by sQTLs and eQTLs. **c**, Colocalization

of sQTLs excising the transmembrane domains of the interleukin receptors *IL6R* and *IL17RA* and mediation with plasma protein quantities ($n = 3,024$ for *IL17RA* and $n = 3,072$ for *IL6R*). The central point represents the mediation effect estimate. Error bars represent the upper and lower 95% confidence intervals of the estimated effects. **d**, Schematic representation of the splicing events excising transmembrane domains of the interleukin receptors *IL6R* and *IL17RA*.

metabolite, comprising 241 significant sQTL mediation models (Supplementary Tables 20 and 21).

Previous reports showed that the missense variant rs2228145 affects IL-6R ectodomain shedding by the alteration of one of the cleavage sites of ADAM10/ADAM17 metalloproteinases[31,32]. In line with this finding, we observed the previously mentioned *IL6R* transmembrane

splicing event mediated a minority of the effect of the lead SNP (rs12126142), which is in high LD ($r^2 > 0.99$; $D' > 0.99$) with this missense variant, on Olink-measured plasma protein abundance (4.67%, $P = 1.12 \times 10^{-4}$; Fig. 4c,d). This suggests a potential dual action of the sSNP or tagged variants on removing this domain and, hence, creating a soluble isoform by both splicing and proteolytic pathways. Conversely,

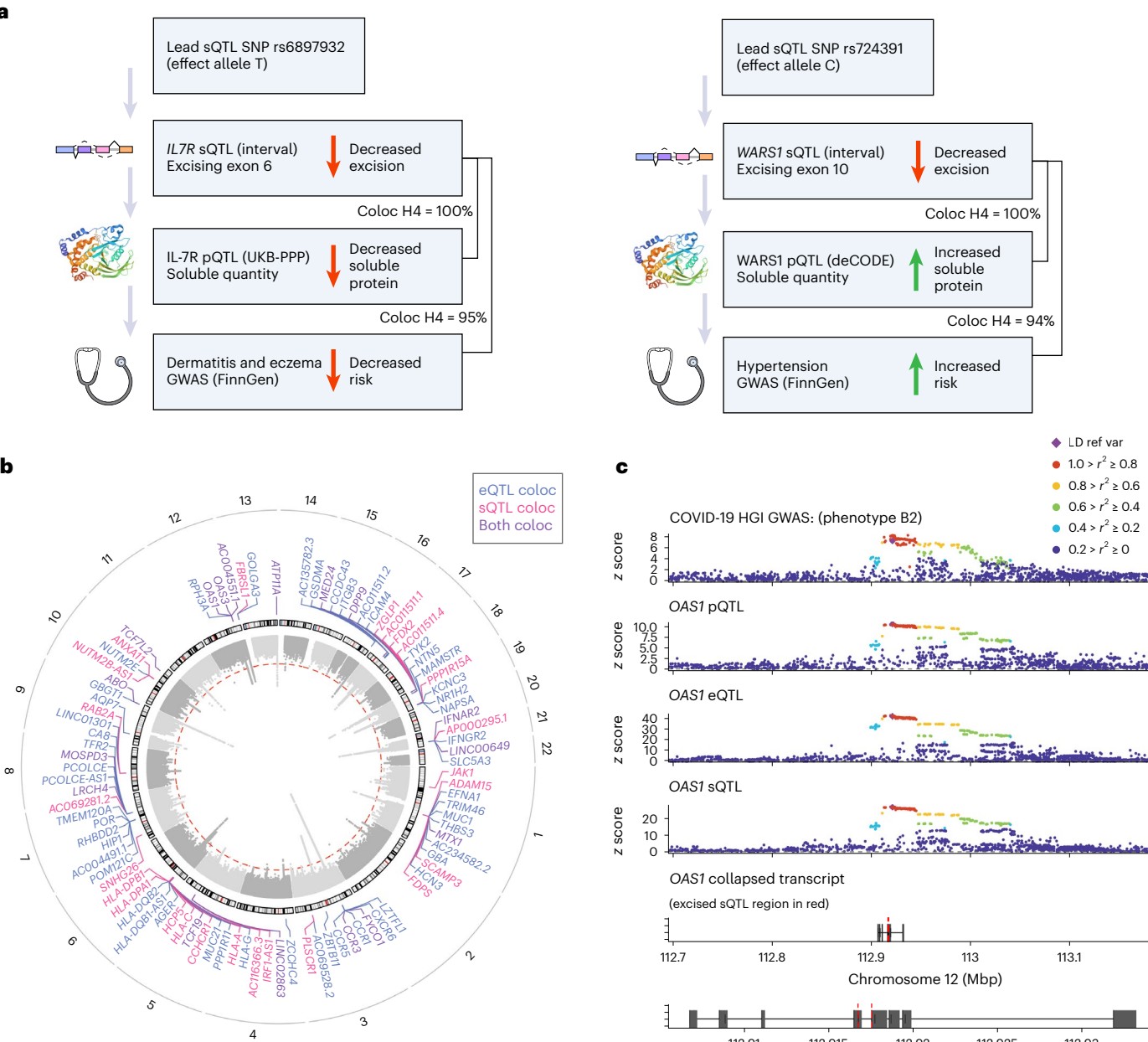

**Fig. 5 | Multitrait colocalization of *cis*-eQTLs and *cis*-sQTLs with molecular phenotypes and health outcomes. a**, Putative pathways and directions of the effect of sQTL signals for *IL7R* and *WARS1* associated with plasma protein quantity, dermatitis and eczema, and hypertension, respectively. The image depicting 'soluble protein' was reproduced from a public database (PDB code 2F6W). **b**, Gene-level summary of colocalization of *cis*-eQTL and

*cis*-sQTL with COVID-19 HGI summary statistics. The red dashed line represents genome-wide significance ($P = 5 \times 10^{-8}$) and the height toward the center represents the significance of the GWAS association. **c**, Example of a multitrait colocalization for COVID-19 in *OAS1*, with GWAS summary statistics, *cis*-pQTL, *cis*-eQTL and *cis*-sQTL.

the colocalized signal (lead *cis*-sSNP rs34495746) between splicing of the transmembrane domain of *IL17RA* and levels of its plasma protein was found to have most of the effect mediated by transcript splicing (90.41%, $P = 1.14 \times 10^{-43}$; Fig. 4c). Consistent with this observation, neither the lead SNP nor any strong tagging SNPs ($r^2 > 0.8$) were missense variants.

**Deconvoluting molecular mechanisms underlying GWAS loci**
Molecular QTLs can provide insights into the mechanisms underlying genetic variants that influence disease risk[33]. We performed colocalization analyses with genetic association signals for 20 disease phenotypes from the FinnGen project (release 9)[34], prioritized based on their

relevance to the circulatory system and available sample size (that is, ≥1,000 cases; Supplementary Table 22).

Disease-associated signals colocalized with 649 *cis*-eGenes and 365 *cis*-sGenes (1,035 splicing events) across all tested traits (Supplementary Tables 23 and 24). Many of these independent signals (136/981 for *cis*-eQTLs and 304/1589 for *cis*-sQTLs) also colocalized with pQTLs and mQTLs, revealing the regulatory pathways underlying the complex trait-associated variants. For example, a *cis*-sQTL for the transmembrane domain splicing of *IL7R* colocalized with an association locus for dermatitis and eczema, as well as a pQTL for IL-7R in UKB-PPP (Fig. 5a). This analysis implicates soluble isoforms of IL-7R generated by alternative splicing in this condition. The alternative

allele of rs6897932 (T) is associated with decreased excision of the *IL7R* transmembrane domain, lower levels of IL-7R in plasma and reduced risk of dermatitis and eczema. This allele has been previously shown to associate with decreased lymphocyte count[35] and decreased risk of multiple sclerosis[36], suggesting consistent therapeutic implications.

Tryptophanyl-tRNA synthetase 1 (encoded by *WARS1*) exists in both secreted and intracellular forms[37], with downstream impacts on vascular permeability[38]. Here we found a *cis*-sQTL for excision of exon 10 of *WARS1* (encoding a portion of the tRNA synthetase protein domain), which colocalized with both the WARS1 pQTLs and risk for hypertension in FinnGen (Fig. 5a). The alternative allele of rs724391 (C) is associated with decreased excision of exon 10, higher plasma protein levels of WARS1 and increased risk of hypertension.

Finally, we also performed a genetic look-up analysis of all independent signals at the identified *cis*-eQTLs and *cis*-sQTLs using data from the Open Target Genetics Portal (v22.10). We present the results on our web portal.

### Transcriptional mechanisms underlying COVID-19 GWAS loci

Most of the whole-blood RNA is derived from circulating immune cells. Given the importance of the host immune response in COVID-19, we conducted colocalization analyses of the identified eQTLs and sQTLs with genetic loci associated with COVID-19 susceptibility and severity available from the pan-biobank COVID-19 Host Genetics Initiative[39]. We found colocalized signals with COVID-19 loci for 67 *cis*-eGenes and 42 *cis*-sGenes (91 splicing events; Supplementary Tables 25 and 26 and Fig. 5b), of which 17 overlapped.

Previous analyses have identified genetic variants that impact splicing of *OAS1* (refs. 40,41). These variants have subsequently been implicated in influencing COVID-19 severity[41]. Consistent with these data, we observed colocalization of an eQTL and sQTLs for seven splicing events at the *OAS1* locus with COVID-19 (Fig. 5c). Adjusting for *OAS1* gene expression levels did not ablate the sQTL signals ($P < 1 \times 10^{-16}$), suggesting the presence of multiple independent transcriptional mechanisms at this locus. In addition, we found colocalization for these eQTLs and sQTLs with the OAS1 pQTL, suggesting that genetic variants mediate disease risk through transcriptional changes impacting soluble protein levels.

Furthermore, the GWAS signals for COVID-19 susceptibility and severity at the *IFNAR2* locus (encoding the interferon α/β receptor 2) colocalized with a *cis*-eQTL, and *cis*-sQTLs associated with 10 splicing events in this gene (Supplementary Fig. 6). This included a splicing event excising exons 8 and 9, encoding the IFNAR2 transmembrane domain. Rare (stop–gain) mutations in exon 9 of this gene leading to loss of function have been previously reported to increase the risk of severe COVID-19 infection[42]. While IFNAR2 was not measured by the proteomic assays, isoforms of *IFNAR2* lacking the transmembrane domain are known to generate a soluble protein isoform[43], and significantly higher quantities of soluble IFNAR2 have been observed in the serum of patients with severe COVID-19 (ref. 44). However, the role of splicing in this gene on disease severity has not been previously reported. Notably, the colocalizing *IFNAR2* eQTLs are also *trans*-sQTLs for five splicing events in *IFI27*, four of which do not have an association in *cis*. Our results provide evidence for a mechanism whereby common variants regulating splicing of *IFNAR2* could be contributing to disease severity through impacts on protein solubility.

## Discussion

Nonprotein-coding genetic variants have an important role in the genetics of complex traits, accounting for 90% of common trait heritability[45]. Genome-wide, multilayered molecular QTL data can help elucidate the functional impact of trait-associated variants and their regulatory networks that underpin complex disease biology. To this end, we discovered eQTLs for 17,233 genes and sQTLs for 29,514 splicing phenotypes in 6,853 genes in peripheral blood through RNA-seq of 4,732

individuals. This included nonprimary signals for 81% of *cis*-eGenes and 49% of *cis*-sGenes, substantially increasing knowledge of the independent genetic determinants of gene expression in whole blood. We combined these data with mQTL and pQTL data in the same participants of the INTERVAL study to map the genetic basis for disease phenotypes. Notably, 52% of independent eQTL colocalizations and 28% of significant mediations involved nonprimary eQTL signals. Similarly, 31% of independent sQTL colocalizations and 30% of significant mediations involved nonprimary sQTL signals. These data demonstrate the value of the conditional analysis performed. Finally, we performed a downsampling analysis to provide guidance toward the expected eQTL and sQTL discovery for future studies (Supplementary Fig. 7a,b).

In comparison to eQTLs, the genetic determinants of splicing have been less thoroughly explored, in particular, how they impact downstream molecular phenotypes and disease risk. Our data support previous findings that splicing QTLs are major contributors to complex traits[46]. Through mapping sQTLs alongside eQTLs, we identified additional independent mechanisms by which genetic variants can influence mRNA and protein levels. For example, the 98 splicing events that colocalized with pQTLs (such as IL-6R and IL-7R) excised protein-coding sequences encoding transmembrane domains. Many of these pQTLs did not colocalize with eQTLs, suggesting that the sQTLs provide the pivotal mechanistic insight, given that genetic effects on splicing are more highly shared between tissues than genetic effects on expression[22]. Furthermore, by identifying and using de novo excision events from the RNA-seq data, we increased the resolution beyond established transcript annotations.

Using the multi-omic data in the INTERVAL study, we systematically performed mediation analyses to evaluate causality in the context of colocalized genetic association signals with molecular traits. In total, we observed 222 molecular phenotypes significantly mediated by gene expression or splicing, providing an additional layer of evidence to delineate functional mechanisms. For instance, we found that an sQTL excising the extracellular domain of *CD33* mediated most of the effect of the sSNP on CD33 soluble protein levels. Mediation analyses are important to define the mode of action of the genetic effects underlying association loci identified in GWAS, as well as the magnitude and direction of their relative effects on downstream phenotypes.

Our study has limitations. First, while we have focused on one definition of transcript splicing due to the annotation-free approach benefiting the downstream analyses, other methodologies may shed light on other aspects of transcript splicing. Second, statistical power was limited to mapping genome-wide eQTLs and sQTLs in *trans*. As *trans*-QTLs are challenging to replicate and distinguish from cell type heterogeneity in bulk RNA-seq studies[9], we prioritized the identified conditionally independent lead *cis*-eSNPs for our *trans*-QTL analyses to prioritize the mechanism of upregulated gene expression modifying the expression and splicing of downstream genes. While we show a strong correlation of effect sizes for SNP–gene pairs also tested by the eQTLGen consortium (phase I), large-scale meta-analyses of *trans*-QTL datasets will be required to create a resource of replicated associations, such as that being prepared by the eQTLGen consortium (phase II; https://www.eqtlgen.org/). To aid in these efforts, we have provided full *trans*-QTL summary statistics on our web portal. Third, our analyses comprised proteins quantified in plasma, rather than intracellular proteins. Thus, the interpretation of the effects of gene expression and splicing QTLs on proteins may be due to impacts on both quantity and solubility of the resulting protein, and other regulatory mechanisms, such as the stability of the mRNA and protein in addition to translational efficiency, may not be captured. However, additional data would be needed to address this. Fourth, the intrinsic properties of the different molecular data types can create challenges in interpretation. For example, there is a considerable correlation structure between metabolite levels[47]. As such, we found that the majority of mQTLs (96%) colocalized with either a *cis*-eQTL or *cis*-sQTL.

Conversely, mQTLs showed mediation by *cis*-eQTLs or *cis*-sQTLs less frequently than pQTLs (that is, 6.8% versus 32.6% for mQTLs and pQTLs, respectively). Finally, our cohort comprised individuals of European ancestry. More work is needed to establish the translatability of our findings to other ancestries.

Previous studies showed that local regulation of gene expression is largely shared across tissues[48] and that larger, well-powered eQTL studies in a surrogate tissue may identify more trait-colocalizing eQTLs than smaller studies in the target tissue[49]. Hence, these results provide a scientific rationale for the generation of increasingly large-scale QTL data in easily accessible tissues, such as peripheral blood. In our study, we further demonstrate the value of such a dataset when integrating data from multiple molecular phenotypes in the same individuals and linking these to external health outcomes to help address the variant-to-function challenge. Similar application to population biobanks is warranted, and with the emerging availability of concomitant molecular data at the single-cell level across a wide range of tissues, single-cell-QTL mapping at the population scale will become feasible. Such data will enable us to dissect gene-regulatory networks at much greater resolution across specific cell types and dynamic processes[50,51]. Together, these improved molecular QTL data will further enhance the interpretation of GWAS signals[52]. While GWAS signals have previously been observed to be depleted for eQTLs[53], we demonstrate that the broader approaches used in this study, such as the increased sample size, the resolution of nonprimary signals and the additional signals captured by the sQTLs, have the potential to increase discovery of the molecular mechanisms underlying GWAS association.

## Online content

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

¹Wellcome Sanger Institute, Wellcome Genome Campus, Hinxton, UK. ²British Heart Foundation Cardiovascular Epidemiology Unit, Department of Public Health and Primary Care, University of Cambridge, Cambridge, UK. ³Victor Phillip Dahdaleh Heart and Lung Research Institute, University of Cambridge, Cambridge, UK. ⁴Cambridge Baker Systems Genomics Initiative, Department of Public Health and Primary Care, University of Cambridge, Cambridge, UK. ⁵Cambridge Baker Systems Genomics Initiative, Baker Heart and Diabetes Institute, Melbourne, Victoria, Australia. ⁶Human Technopole, Fondazione Human Technopole, Milan, Italy. ⁷Australian Centre for Precision Health, Unit of Clinical Health Sciences, University of South Australia, Adelaide, South Australia, Australia. ⁸South Australian Health and Medical Research Institute, Adelaide, South Australia, Australia. ⁹Department of Epidemiology, Harvard T.H. Chan School of Public Health, Boston, MA, USA. ¹⁰Health Data Research UK Cambridge, Wellcome Genome Campus and University of Cambridge, Cambridge, UK. ¹¹British Heart Foundation Centre of Research Excellence, University of Cambridge, Cambridge, UK. ¹²Translational Sciences, Research & Development, Biogen, Cambridge, MA, USA. ¹³Neuroscience Data Science, Janssen Research & Development, Cambridge, MA, USA. ¹⁴Centre for Genomics Research, Discovery Sciences, BioPharmaceuticals R&D, AstraZeneca, Cambridge, UK. ¹⁵Department of Medicine, University of Melbourne, Austin Health, Melbourne, Victoria, Australia. ¹⁶Genomics, BioMarin Pharmaceutical Inc., Novato, CA, USA. ¹⁷Radcliffe Department of Medicine, John Radcliffe Hospital, Oxford, UK. ¹⁸Clinical Services, NHS Blood and Transplant, Oxford Centre, John Radcliffe Hospital, Oxford, UK. ¹⁹NIHR Blood and Transplant Research Unit in Donor Health and Behaviour, University of Cambridge, Cambridge, UK. ²⁰Department of Immunology and Inflammation, Imperial College London, London, UK. ²¹These authors contributed equally: Alex Tokolyi, Elodie Persyn, Michael Inouye, Emma E. Davenport, Dirk S. Paul. ✉e-mail: dirk.paul@astrazeneca.com

## Methods

### Study participants

The INTERVAL study is a prospective cohort study of approximately 50,000 participants nested within a randomized trial of varying blood donation intervals[15,16]. Between 2012 and 2014, blood donors aged 18 years and older were recruited at 25 centers of England's National Health Service Blood and Transplant (NHSBT). All participants gave written informed consent before joining the study, and the National Research Ethics Service approved this study (11/EE/0538). Participants were generally in good health, as blood donation criteria exclude individuals with a history of major diseases (for example, myocardial infarction, stroke, cancer, HIV and hepatitis B or C) and who have had a recent illness or infection. Participants completed an online questionnaire comprising questions on demographic characteristics (for example, age, sex and ethnicity), lifestyle (for example, alcohol and tobacco consumption), self-reported height and weight, diet and use of medications. Demographic details are provided in Supplementary Table 27.

### Blood collection

Blood samples were collected from all INTERVAL participants at baseline and also from ~60% of participants ~24 months after baseline. For a subset of ~5,000 participants at the 24-month time point, an aliquot of 3 ml of whole blood was collected in Tempus Blood RNA Tubes (Thermo Fisher Scientific), following the manufacturer's instructions, and then transferred at ambient temperature to the UK Biocentre. Samples were stored at −80 °C until use.

### RNA extraction

RNA extraction was performed by QIAGEN Genomic Services using QIAGEN's proprietary silica technology. The quality control (QC) of the extracted RNA was performed by spectrophotometric measurement on an Infinite 200 Microplate Reader (Tecan). RNA integrity number (RIN) values were determined using a TapeStation 4200 system (Agilent), following the manufacturer's protocol. Samples with a concentration <20 ng μl$^{-1}$ and a RIN value <4 were excluded from further analyses.

### Automated RNA-seq library preparation

Samples were quantified with a QuantiFluor RNA System (Promega) using a Mosquito LV liquid handling platform (SPT Labtech), Bravo automation system (Agilent) and FLUOstar Omega plate reader (BMG Labtech) and then cherry-picked to 200 ng in 50 μl (=4 ng μl$^{-1}$) using a liquid handling platform (Tecan Freedom EVO). Next, mRNA was isolated using a NEBNext Poly(A) mRNA Magnetic Isolation Module (New England Biosciences, NEB) and then resuspended in nuclease-free water. Globin depletion was performed using a KAPA RiboErase Globin Kit (Roche). RNA library preparation was done using a NEBNext Ultra II RNA Library Prep Kit for Illumina (NEB) on a Bravo NGS workstation automation system (Agilent). PCR was performed using a KapaHiFi HotStart ReadyMix (Roche) and unique dual-indexed tag barcodes on a Bravo NGS workstation automation system (Agilent). We applied the following PCR program: 45 s at 98 °C, 14 cycles of 15 s at 98 °C, 30 s at 65 °C and 30 s at 72 °C, followed by 60 s at 72 °C. Using a Zephyr liquid handling platform (PerkinElmer), PCR products were purified using AMPure XP SPRI beads (Agencourt) at a 0.8:1 bead-to-sample ratio and then eluted in 20 μl of elution buffer (Qiagen). RNA-seq libraries were quantified with an AccuClear Ultra High Sensitivity dsDNA Quantitation Kit (Biotium) using a Mosquito LV liquid handling platform (SPT Labtech), Bravo automation system (Agilent) and FLUOstar Omega plate reader (BMG Labtech). Then, libraries were pooled up to 95-plex in equimolar amounts on a Biomek NX-8 liquid handling platform (Beckman Coulter), quantified using a High Sensitivity DNA Kit on a 2100 Bioanalyzer (Agilent) and then normalized to 2.8 nM before sequencing.

### RNA-seq and data preprocessing

Samples were sequenced using 75 bp paired-end sequencing reads (reverse stranded) on a NovaSeq 6000 system (S4 flow cell, Xp workflow; Illumina). The sequencing data were deplexed into separate compressed reference-oriented alignment map (CRAM) files for each library in a lane. Adapters that had been hard-clipped before alignment were reinserted as soft-clipped postalignment, and duplicated fragments were marked in the CRAM files. The data preprocessing, including sequence QC and STAR alignments, was performed with the Nextflow pipeline publicly available at https://github.com/wtsi-hgi/nextflow-pipelines/blob/rna_seq_interval_5591/pipelines/rna_seq.nf, including the specific aligner parameters. We assessed the sequence data quality using FastQC (v0.11.8). Samples mismatched between RNA-seq and genotyping data within the cohort were identified using QTLtools MBV (v1.2)[54]. Reads were aligned to the GRCh38 human reference genome (Ensembl GTF annotation v99) using STAR (v2.7.3a)[55]. The STAR index was built against GRCh38 Ensembl GTF v99 using the option -jdbOverhang 75. STAR was run in a two-pass setup with standard ENCODE options to increase mapping accuracy: (1) a first alignment step of all samples was used to discover new splice junctions, (2) splice junctions of all samples from the first step were collected and merged into a single list, (3) a second step realigned all samples using the merged splice junctions list as input. We used featureCounts (v2.0.0)[56] to obtain a count matrix.

### QC of gene expression data

Sequencing was performed across 15 batches. We filtered samples of poor quality by removing samples with a read depth below 10 million uniquely mapped reads. On average, each sample had 25.3 million unique reads (interquartile range = 21.5–26.9, including batches 1 and 15 for which libraries were sequenced twice). A relatedness matrix was obtained using the PLINK (v1.9)[57] -make-rel 'square' command on pruned genotype data, and a cutoff threshold of 0.1 was used to define related individuals. For each pair of related individuals, one individual was arbitrarily removed. After QC, a total of $n$ = 46 samples were removed. After the sample QC, we filtered lowly expressed genes by retaining genes with >0.5 counts per million (CPM) in ≥1% of the samples, in line with the filter applied by the eQTLGen consortium[9]. In our dataset, a CPM value of 0.5 roughly equates to having 5 counts in a sample with the lowest read depth (10 million uniquely mapped reads) or 47 counts in a sample with the highest read depth (94 million reads). We further excluded globin genes, rRNA genes and pseudogenes. After QC, the final gene expression dataset included 19,173 autosomal genes (13,874 of which are protein-coding) across a total of 4,732 individuals.

### Normalization of gene expression data

Before the eQTL analysis, the count data were normalized using the trimmed mean of $M$ values (TMM)[58] implemented in the R package edgeR (v3.24.3). The TMM-normalized values were further converted into fragments per kilobase of transcript per million mapped reads (FPKM) values (log$_2$-transformed) to take gene length into account. Next, for each gene, the normalized log$_2$-FPKM values across samples were transformed via the ranked-based inverse normal transformation function 'rntransform' implemented in the R package GenABEL (v1.8-0)[59]. Inverse normal transformation was applied to ensure the expression values followed a normal distribution.

### Splicing data generation

Splice junctions were extracted from aligned RNA-seq BAMs for the 4,732 individuals using Regtools (v0.5.2)[60] junctions extract (parameters: '-s 1 -m 50'). Introns represented by extracted splice junctions were then clustered into groups based on overlapping start or end sites, with the Leafcutter pipeline (v0.2.9)[52] (leafcutter_cluster_regtools.py, parameters: '-m 100 -M 50 -l 100000 -p 0.01'). Clustered introns were then prepared for sQTL analysis with Leafcutter

prepare_phenotype_table.py to convert intron counts to normalized ratios and compute ten splicing principal components (PCs). Introns were matched to regions of Ensembl v99 genes and protein domains annotated with R v4.0.3 using a custom pipeline (described in Data availability). Total observed introns (*n* = 956,722) were filtered to those that were autosomal, overlapping an expressed gene body, with CPM > 0.5 in at least 24 individuals, and sufficient variance (minimum two filtered splice event phenotypes per cluster), resulting in 111,937 filtered splicing event phenotypes, in 11,016 genes (see Supplementary Fig. 8 for a summary of splicing event QC).

### DNA extraction, genotyping and imputation

In brief, DNA extracted from buffy coat samples collected from INTER-VAL participants at the study baseline was used to assay approximately 830,000 variants on the Affymetrix Axiom UK Biobank genotyping array[61]. Genotyping and sample QC were performed as previously described[61]. Before imputation, additional variant filtering steps were performed to establish a high-quality imputation scaffold, including 654,966 autosomal, nonmonomorphic, bi-allelic variants with Hardy–Weinberg equilibrium (HWE) $P > 5 \times 10^{-6}$, with a call rate of >99% across the INTERVAL genotyping batches in which a variant passed QC and a global call rate of >75% across all INTERVAL genotyping batches. Next, variants were phased using SHAPEIT3 and imputed using a combined 1000 Genomes Phase 3-UK10K reference panel. Imputation was per-formed via the Sanger Imputation Server (https://imputation.sanger.ac.uk) and resulted in 87,696,888 imputed variants. For the present analysis, imputed genotypes were lifted over to reference build GRCh38 using CrossMap (v0.3.4)[62] and the Ensembl chain file provided with the package. Imputed genotypes were hard-called with PLINK (v2.00a2-32-bit)[57] using the default parameters. Before analysis, the dataset was restricted to individuals with RNA-seq and filtered to remove genetic variants with HWE exact test $P < 1 \times 10^{-6}$, genotype missingness >0.05 or minor allele frequency < 0.5%.

### Identification of sample swaps and cross-contamination

The Match BAM to VCF (MBV) method from QTLTools[54] was used to identify sample mix-ups and cross-contamination. MBV directly com-pares each aligned RNA-seq BAM file to all the genotypes in the VCF file and computes the proportion of concordant heterozygous and homozygous sites. To reduce computation time, we only focused on chromosome 1. Based on the concordance (close to 100%) between the genotype data and RNA-seq samples, we identified and corrected ten pairs of mislabeled samples. We removed seven RNA-seq samples that did not show a clear high concordance (the highest was <50%) with any particular genotype sample—either due to cross-contamination or the actual matching genotypes were not available.

### PEER factor and splicing PC analysis

We used the probabilistic estimation of expression residuals (PEER) method[63], implemented in the R package peer v.1.0 (downloaded from https://github.com/PMBio/peer), to detect and correct eQTL mapping for latent batch effects and other unknown confounders. PEER factors were estimated while accounting for age, sex, body mass index and 19 blood cell traits (Supplementary Table 28) as known confounders. PEER was run for 50 factors, converging at 148 itera-tions. For inclusion in the eQTL analysis, we selected the number of PEER factors based on the following two criteria: (1) discovery of the largest number of *cis*-eGenes and (2) additional gain in *cis*-eGenes with incremental increase in PEER factors (Supplementary Fig. 9a). We found that the relationship between the increase in the number of discovered *cis*-eGenes and the incremental increase in PEER factors is similar to that observed in the GTEx whole-blood dataset[22]. Therefore, we included 35 PEER factors in our eQTL analysis, consistent with GTEx. We used a similar approach to determine the optimal number of PCs to include in the sQTL analysis, testing 0–10 PCs. We found that *cis*-sQTL

discovery only increased slightly with the number of PCs included with no obvious threshold (Supplementary Fig. 9b). Given that ten PCs were established as a previous default for sQTL mapping[52], we opted to include ten splicing PCs.

### Mapping of eQTLs and sQTLs

eQTLs and splicing QTLs were called using tensorQTL (v1.0.6)[64] and postprocessed with a custom pipeline[65]. The covariates integrated into the regression model are listed and described in Supplementary Tables 27 and 28. In brief, these included (1) demographic variables such as age at blood sampling, sex and body mass index at baseline (because it was not collected at the time of blood sampling), (2) tech-nical variables such as RIN, read depth and season of blood sampling, (3) ten genotype PCs and 35 PEER factors (for eQTLs) or ten splicing PCs (for sQTLs) and (4) 19 different blood cell traits. For the *cis*-eQTL analysis, variants were defined as being in *cis* with a gene if they were located within a window of ±1 Mb from the TSS. For the sQTL analysis, the window was set to ±500 kb from the center of the splicing event to balance primary and secondary sQTL discovery. Feature annota-tion, including TSS position, was obtained from Ensembl v99 (Janu-ary 2020). For both *cis*-eQTL and *cis*-sQTL analyses, multiple-testing correction was applied in tensorQTL as follows: (1) for each gene (or splicing event), the adjusted lowest *P* value was estimated using a β distribution approximation from a permutation procedure (10,000 permutations)[66]; (2) Benjamini–Hochberg FDR correction was applied to the β-approximated *P* values across genes (or splicing events), and the FDR *q* value threshold was set to 5%. For each significant gene (or splicing event), a nominal *P*-value threshold was estimated to identify significant SNPs. To demonstrate how increased sample size assists in *cis*-eQTL and *cis*-sQTL discovery, random samples of patients in *n* = 500 increments were subsetted and QTL mapping was performed with the same inputs as the full *cis*-QTL analyses, with the output being the number of significant genes (or splice phenotypes) with a significant eQTL (or sQTL). Conditional analysis was performed for each *cis*-eGene (or splicing event phenotype with a *cis*-sQTL) using GCTA-COJO v1.94.0beta (January 2022)[67,68]. The program took as input the gene *cis*-eQTL (or *cis*-sQTL) summary statistics, the INTERVAL imputed genotype data for *cis*-variants and the *P*-value threshold used to identify the *cis*-eGene (or splicing QTL). A *trans*-eQTL analysis was performed on the list of lead SNPs from *cis*-eGenes independent sig-nals. The *trans*-regions were defined as genomic regions outside of the ±5 Mb window from the TSS. The Bonferroni multiple-testing cor-rection method (that is, $P = 0.05$/number of tested *trans*-associations) was applied to identify significant *trans*-associations. While previous work has demonstrated that *trans*-QTL analyses may be susceptible to artifacts due to read cross-mapping between similar genes[69], our quantification approach using only uniquely mapped reads led to only a small fraction of our *trans*-QTL results (that is, 12.2% of *trans*-eQTLs and 23.6% of *trans*-sQTLs) involving genes that were flagged for sequence similarity. For the *trans*-QTL analyses, we also assessed if there were PEER factors or splicing PCs associated with *cis*-eSNPs. We did not detect any significant associations ($P < 9.4 \times 10^{-8}$, Bon-ferroni multiple-testing correction across 53,457 *cis*-eSNPs and ten splicing PCs) between *cis*-eSNPs and splicing PCs. However, we found that ten *cis*-eSNPs were significantly associated with five PEER factors ($P < 2.7 \times 10^{-8}$, Bonferroni multiple-testing correction across 53,457 *cis*-eSNPs and 35 PEER factors). As a sensitivity analysis, we performed the *trans*-eQTL analyses with and without these five PEER factors. We showed a high correlation of *z* scores for *trans*-eQTL involving these ten SNPs before and after removing these PEER factors from the model (Pearson correlation = 0.80; Supplementary Fig. 10). We identified sig-nificant associations of *cis*-eSNPs with an additional 121 *trans*-eGenes. Overall, we identified a higher number of *trans*-eGenes by integrating all 35 PEER factors in the model (that is, 2,058 *trans*-eGenes instead of 1,811 *trans*-eGenes).

## Validation of *cis*-eQTL and *cis*-sQTL results

Results from the *cis*-eQTL analysis were compared to the results obtained in the eQTLGen study[9], which are available at https://www.eqtlgen.org/cis-eqtls.html. In our comparison, we explored the percentage of overlap of *cis*-eGenes and the effect direction of genetic associations. For the overlap of *cis*-eGenes, we focused on the list of 15,722 genes that were tested in both INTERVAL and eQTLGen. For the comparison of effect directions, we computed the correlation of $z$ scores for SNPs that were the most significant in INTERVAL for each gene and that were also tested in eQTLGen. Results from sQTL analysis were collapsed to the sGene level for comparison to GTEx whole-blood sQTLs (v.8)[22], which are available at https://gtexportal.org/home/datasets.

## Enrichment analyses

Enrichment analyses were performed using a one-sided Fisher's exact test on QTL results annotated with GO terms[70] (downloaded in May 2022) and the Human Transcription Factors database[23]. We tested for enrichment within *cis*-eGenes with a *trans*-association with gene expression or splicing using significant *cis*-eGenes as background. Benjamini–Hochberg FDR correction was applied to identify significant enrichment.

## Colocalization analysis

Colocalization analysis was performed using the results of conditional analysis from GCTA-COJO[67,68] and the R package Coloc (v5.1.0.1)[71] on pairwise independent QTL signals following the pwCoCo methodology[72]. The colocalization analysis window was the entire *cis*-window, that is, ±1 Mb for eQTLs and ±500 kb for sQTLs. Prior probabilities were kept as the default values, that is, $P_1 = 1 \times 10^{-4}$, $P_2 = 1 \times 10^{-4}$ and $P_{12} = 1 \times 10^{-5}$. Colocalized results were defined with the thresholds PP3 + PP4 ≥ 0.9 and PP4/PP3 ≥ 3, PP3 and PP4 being the posterior probabilities of hypotheses 3 and 4 as outlined previously[71]. For colocalization analysis with external omics data, summary statistics were downloaded from each study (see Supplementary Table 29 for the description of the different omics studies). A previous study performed simulations showing that the impact of complete sample overlap on colocalization results was negligible with 200 individuals and, therefore, will be even smaller with large sample sizes as used here[73]. Before colocalization analysis, (1) proteins were annotated using the R package biomaRt (v2.46.3) to obtain corresponding genes in Ensembl v99 (January 2020), and (2) significant pQTLs and mQTL were filtered. For pQTLs, $P$-value thresholds per feature were defined by a two-step multiple-testing correction[74,75]. For mQTLs, we used a Bonferroni-adjusted $P$-value threshold of $P < 5 \times 10^{-8}$, corrected for the number of metabolites analyzed.

## Mediation analysis

Mediation analyses were conducted using the natural effects model implemented in the R package Medflex (v0.6-7)[76]. In the models, we defined (1) the independent lead *cis*-eQTL (or *cis*-sQTL) SNP (coded as 0, 1 and 2) as the independent (exposure) variable, (2) the gene expression level (or splicing event phenotype) of the *cis*-eGene/-sGene as the mediator and (3) the molecular trait as the dependent (outcome) variable. Gene expression (or splicing event phenotype) residuals were computed after adjusting for the same covariates as we used for eQTL/sQTL mapping, while molecular traits were adjusted for covariates described by each study (Supplementary Table 29). For all mediation analyses, samples with missing genotype or molecular data were removed. Standard errors were computed based on the robust sandwich estimator. Significant direct, indirect and total effects were identified after Bonferroni multiple-testing correction between each molecular phenotype assay.

## Interactive QTL web portal

To facilitate the accessibility of the results, a web portal was built to enable the exploration of eQTL and sQTLs. Summary statistics and expression phenotypes were imported into a MariaDB (v10.2.38) database; code was written to facilitate their retrieval in PHP (v7.2.34) with jQuery (v3.5.1) and styled with Bootstrap (v3.4.1). Tables are powered by DataTables (v1.13.3). Locus and QTL plots are visualized with LocusZoomJS (v0.13.4) and plotly (v2.9.0), respectively.

## Statistics and reproducibility

No statistical method was used to predetermine the sample size. No data were excluded from the analyses, except due to the QC steps detailed above. The experiments were not randomized, and the investigators were not blinded to allocation during experiments and outcome assessment.

## Reporting summary

Further information on research design is available in the Nature Portfolio Reporting Summary linked to this article.

## Data availability

The INTERVAL study data used in this paper are available to bona fide researchers from ceu-dataaccess@medschl.cam.ac.uk. The data access policy for the data has been approved by the ethics committee and is available at https://www.donorhealth-btru.nihr.ac.uk/ wp-content/uploads/2020/04/Data-Access-Policy-v1.0-14Apr2020.pdf. The release of data is regulated by the Blood Donors Studies BioResource Data Access Committee (DAC). The DAC will review the project's scientific excellence and alignment of the proposal with the overall aims of the database; the research team's experience and capability to conduct the proposed study; and the suitability of the data and any risk to participant confidentiality. The data access process takes approximately 2 months. The newly generated RNA-seq data ($n = 4,732$ INTERVAL participants) have been deposited at the European Genome-phenome Archive under the accession EGAD00001008015. The results from the genetic association, colocalization and mediation analyses are available at https://IntervalRNA.org.uk. The summary statistics are also made available on the above web portal, as well as mirrored on Zenodo (https://doi.org/10.1101/2023.11.25.23299014)[77]. Our data used annotation from Ensembl (https://www.ensembl.org/). For enrichment analyses, we used the public databases GO (https://geneontology.org/)[70] and The Human Transcription Factors (https://humantfs.ccbr.utoronto.ca/)[23].

## Code availability

All original code has been deposited on GitHub (https://github.com/INTERVAL-RNAseq/manuscript-scripts) and a static version archived on Zenodo (https://doi.org/10.5281/zenodo.14015194)[65].

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

## Acknowledgements

Participants in the INTERVAL randomized controlled trial were recruited with the active collaboration of NHSBT England (https://www.nhsbt.nhs.uk/), which has supported fieldwork and other elements of the trial. DNA extraction and genotyping were cofunded by the National Institute for Health and Care Research (NIHR), the NIHR BioResource (https://bioresource.nihr.ac.uk/) and the NIHR Cambridge Biomedical Research Centre (BRC-1215-20014). RNA-seq was funded as part of an alliance between the University of Cambridge and the AstraZeneca Centre for Genomics Research, as well as by the NIHR Cambridge Biomedical Research Centre (BRC-1215-20014). Olink Target assays (Neurology panel) were funded by Biogen. SomaLogic assays were funded by Merck & Co and the NIHR Cambridge Biomedical Research Centre (BRC-1215-20014). Metabolon HD4 assays were funded by the NIHR BioResource, NIHR Cambridge Biomedical Research Centre (BRC-1215-20014), Wellcome Trust (number 206194) and BioMarin Pharmaceutical. Nightingale Health assays were funded by the European Commission Framework Programme 7 (HEALTH-F2-2012-279233). The academic coordinating center for INTERVAL was supported by core funding from the NIHR Blood and Transplant Research Unit (BTRU) in Donor Health and Genomics (NIHR BTRU-2014-10024), NIHR BTRU in Donor Health and Behavior (NIHR203337), UK Medical Research Council (MR/L003120/1), British Heart Foundation (SP/09/002, RG/13/13/30194, RG/18/13/33946 and RG/F/23/110103), BHF Chair Award (CH/12/2/29428) and NIHR Cambridge BRC (BRC-1215-20014 and NIHR203312). A complete list of the investigators and contributors to the INTERVAL trial is provided in ref. 16. The academic coordinating center thanks blood donor center staff and blood donors for participating in the INTERVAL trial. This work was supported by Health Data Research UK, which is funded by the UK Medical Research Council, Engineering and Physical Sciences Research Council, Economic and Social Research Council, Department of Health and Social Care (England), Chief Scientist Office of the Scottish Government Health and Social Care Directorates, Health and Social Care Research and Development Division (Welsh Government), Public Health Agency (Northern Ireland), British Heart Foundation and Wellcome. The views expressed are those of the authors and not necessarily those of the NIHR or the Department of Health and Social Care. The Wellcome Sanger Institute is supported by core funding from the Wellcome Trust (206194 and 220540/Z/20/A). We thank the Wellcome Sanger Institute's Scientific Operations team for their contribution to sequencing data generation. For Open Access, the authors have applied a CC BY public copyright license to any Author Accepted Manuscript version arising from this submission. This work was performed using resources provided by the Cambridge Service for Data Driven Discovery (CSD3) operated by the University of Cambridge Research Computing Service (https://www.csd3.cam.ac.uk/), provided by Dell EMC and Intel using Tier-2 funding from the Engineering and Physical Sciences Research Council (capital grant EP/P020259/1), and DiRAC funding from the Science and Technology Facilities Council (https://dirac.ac.uk/). We thank the participants and investigators of the UK Biobank study who made this work possible (resource applications 26041 and 65851). A.T. is supported by the Wellcome Trust (PhD studentship 222548/Z/21/Z). E.P. is funded by the EU/EFPIA Innovative Medicines Initiative Joint Undertaking BigData@Heart (grant 116074) and by the NIHR BTRU in Donor Health and Behavior (NIHR203337). S.C.R. is funded by a BHF Programme Grant (RG/18/13/33946) and the NIHR Cambridge BRC (BRC-1215-20014 and NIHR203312). Y.X. is supported by the UK Economic and Social Research Council (ES/T013192/1). J.E.P. is supported by a Medical Research Foundation Fellowship (MRF-057-0003-RG-PETE-C0799). J.D. holds a British Heart Foundation Professorship and a NIHR Senior Investigator Award. M.I. is supported by the Munz Chair of Cardiovascular Prediction and Prevention and the NIHR Cambridge Biomedical Research Centre (BRC-1215-20014 and NIHR203312), and is also supported by the UK Economic and Social Research Council (ES/T013192/1).

## Author contributions

A.T., E.P., M.I., E.E.D. and D.S.P. conceived the study design and wrote the paper. A.T. and E.P. conducted statistical analyses. A.P.N. performed the QC of RNA-seq data. G.N. and V.I. performed the preprocessing of RNA-seq data. J.M. made substantial contributions to the QTL mapping. A.T. developed the interactive web portal. K.L.B., T.V., M.T., D.S., X.J., S.A.L., S.C.R., Y.X., J.E.P. and A.S.B. provided critical comments on the paper. B.F., M.A.Q., D.R. and S.A.J.T. performed

RNA-seq experiments. B.B.S., C.D.W., H.R., S.P., D.J.G., D.J.R., E.D.A., N.S. and J.D. provided materials and data resources. K.L.B. and A.S.B. provided critical suggestions to the study design. M.I., E.E.D. and D.S.P. supervised the project. All authors reviewed and approved the final version of the paper.

## Competing interests

M.A.Q. is on the Key Opinion Leader panel for New England Biolabs. B.B.S. and H.R. are employees and stockholders of Biogen. C.D.W. is an employee and stockholder of Johnson & Johnson. S.P. and D.S.P. are employees and stockholders of AstraZeneca. D.J.G. is an employee and stockholder of BioMarin Pharmaceutical. D.J.R. is an employee of NHSBT. J.E.P. has received hospitality and travel expenses to speak at Olink-sponsored academic meetings (none within the past 5 years). A.S.B. has received grants outside of this work from AstraZeneca, Bayer, Biogen, BioMarin and Sanofi. M.I. is a trustee of the Public Health Genomics Foundation, a member of the Scientific Advisory Board of Open Targets and has a research collaboration with AstraZeneca that is unrelated to this study. The other authors declare no competing interests.

## Additional information

**Correspondence and requests for materials** should be addressed to Dirk S. Paul.

# Reporting Summary

## Statistics

For all statistical analyses, confirm that the following items are present in the figure legend, table legend, main text, or Methods section.

| n/a | Confirmed | |
|---|---|---|
| ☐ | ☒ | The exact sample size (*n*) for each experimental group/condition, given as a discrete number and unit of measurement |
| ☐ | ☒ | A statement on whether measurements were taken from distinct samples or whether the same sample was measured repeatedly |
| ☐ | ☒ | The statistical test(s) used AND whether they are one- or two-sided<br>*Only common tests should be described solely by name; describe more complex techniques in the Methods section.* |
| ☐ | ☒ | A description of all covariates tested |
| ☐ | ☒ | A description of any assumptions or corrections, such as tests of normality and adjustment for multiple comparisons |
| ☐ | ☒ | A full description of the statistical parameters including central tendency (e.g. means) or other basic estimates (e.g. regression coefficient) AND variation (e.g. standard deviation) or associated estimates of uncertainty (e.g. confidence intervals) |
| ☐ | ☒ | For null hypothesis testing, the test statistic (e.g. *F*, *t*, *r*) with confidence intervals, effect sizes, degrees of freedom and *P* value noted<br>*Give P values as exact values whenever suitable.* |
| ☐ | ☒ | For Bayesian analysis, information on the choice of priors and Markov chain Monte Carlo settings |
| ☒ | ☐ | For hierarchical and complex designs, identification of the appropriate level for tests and full reporting of outcomes |
| ☐ | ☒ | Estimates of effect sizes (e.g. Cohen's *d*, Pearson's *r*), indicating how they were calculated |

*Our web collection on statistics for biologists contains articles on many of the points above.*

## Software and code

Policy information about availability of computer code

| | |
|---|---|
| Data collection | Genotyping and RNA-seq data collection was performed using the standard vendor bioinformatic pipeline for the Affymetrix Axiom UK Biobank genotyping array and the NovaSeq 6000 respectively. Details for the collection of previously generated protein and metabolite data are listed in Supplementary Table 29, and associated manuscripts. Covariate and population meta-data were previously collected as described in 10.1016/S0140-6736(17)31928-1. |
| Data analysis | Software: CrossMap v0.3.4, FastQC v0.11.8,  featureCounts v2.0.0, GCTA-COJO v1.94.0beta (January 2022), Leafcutter v0.2.9, PLINK v1.9, PLINK v2.00a2-32-bit, QTLtools MBV v1.2, R v4.0.3, regtools v0.5.2, SHAPEIT3, STAR v2.7.3a, tensorQTL v1.0.6.<br>R packages: biomaRt v2.46.3, coloc v5.1.0.1, edgeR v3.24.3, GenABEL v1.8-0, medflex v0.6-7, peer v.1.0.<br>Interactive QTL web-portal: Bootstrap v3.4.1, DataTables v1.13.3, jquery v3.5.1, LocusZoomJS v0.13.4, MariaDB v10.2.38, PHP v7.2.34, plotly v2.9.0.<br>Further information about the software and R packages used for data analyses are provided in the Methods section of the manuscript. The main code for analyses is available at https://github.com/INTERVAL-RNAseq/manuscript-scripts. |

For manuscripts utilizing custom algorithms or software that are central to the research but not yet described in published literature, software must be made available to editors and reviewers. We strongly encourage code deposition in a community repository (e.g. GitHub). See the Nature Portfolio guidelines for submitting code & software for further information.

## Data

Policy information about <u>availability of data</u>

All manuscripts must include a <u>data availability statement</u>. This statement should provide the following information, where applicable:
- Accession codes, unique identifiers, or web links for publicly available datasets
- A description of any restrictions on data availability
- For clinical datasets or third party data, please ensure that the statement adheres to our <u>policy</u>

The INTERVAL study data used in this paper are available to bona fide researchers from ceu-dataaccess@medschl.cam.ac.uk. The data access policy for the data is available at http://www.donorhealth-btru.nihr.ac.uk/project/bioresource. The newly generated RNA-sequencing data (n=4,732 INTERVAL participants) have been deposited at the European Genome-phenome Archive (EGA) under the accession number EGAD00001008015. The results from the genetic association, colocalization and mediation analyses are available in the Supplementary Tables, and online at https://IntervalRNA.org.uk. The full summary statistics are also made available on the above web portal, as well as mirrored on Zenodo (10354433). For external and previously computed summary statistics for proteins and metabolites, these accession details are listed in Supplementary Table 29 and associated manuscripts. GTEX v8 summary statistics were downloaded from the online repository (https://gtexportal.org/), and eQTLGen (Phase 1) from their repository (https://www.eqtlgen.org/). The GRCh38 reference genome was sourced from NCBI (https://www.ncbi.nlm.nih.gov/datasets/genome/GCF_000001405.26/) and annotations from the Ensembl V99 GTF (https://www.ensembl.org/). For enrichment analyses, we used the public databases Gene Ontology (https://geneontology.org/) and the Human Transcription Factors database (https://humantfs.ccbr.utoronto.ca/).

## Research involving human participants, their data, or biological material

Policy information about studies with <u>human participants or human data</u>. See also policy information about <u>sex, gender (identity/presentation), and sexual orientation</u> and <u>race, ethnicity and racism</u>.

| | |
|---|---|
| Reporting on sex and gender | Study participants were recruited as part of the INTERVAL study. For the analysis of the RNA-sequencing data, there were 2,105 female and 2,627 male participants. |
| Reporting on race, ethnicity, or other socially relevant groupings | Participants from the INTERVAL study were recruited across England, UK. After quality control of the genotype data, remaining participants were from European ancestry only. No prior ethnicity information was used for filtering participants. |
| Population characteristics | Participants were generally in good health as blood donation criteria excluded individuals with a history of major diseases (e.g., myocardial infarction, stroke, cancer, HIV, and hepatitis B or C) and who had a recent illness or infection. Participant age ranged from 20-79, with a median of 58. Additional population demographic characteristics are described in Supplementary Table 27. |
| Recruitment | Between 2012 and 2014, blood donors aged 18 years and older were recruited at 25 centers of England's National Health Service Blood and Transplant (NHSBT). |
| Ethics oversight | All participants gave informed consent before joining the study and the National Research Ethics Service approved this study (11/EE/0538). |

Note that full information on the approval of the study protocol must also be provided in the manuscript.

# Field-specific reporting

Please select the one below that is the best fit for your research. If you are not sure, read the appropriate sections before making your selection.

☒ Life sciences ☐ Behavioural & social sciences ☐ Ecological, evolutionary & environmental sciences

For a reference copy of the document with all sections, see nature.com/documents/nr-reporting-summary-flat.pdf

# Life sciences study design

All studies must disclose on these points even when the disclosure is negative.

| | |
|---|---|
| Sample size | Raw RNA-sequencing data included 4,778 individuals. After quality control, the RNA-sequencing data included 4,732 individuals. No methodology was used to predetermine sample size, though this study is more than 7 times larger than previous similar studies. Sample sizes of external datasets that were integrated in the different analyses are described in the Supplementary Table 29. |
| Data exclusions | We filtered samples of poor quality by removing samples with a read depth below 10 million uniquely mapped reads. A relatedness matrix was obtained using the PLINK v1.9 -make-rel 'square' command on pruned genotype data, and a cut-off threshold of 0.1 was used to define related individuals. For each pair of related individuals, one individual was arbitrarily removed. After quality control, a total of N=46 samples were removed. |
| Replication | We did not replicate our eQTL and sQTL mapping in a separate dataset but validated our results by performing a comparison with results obtained from external studies (i.e., eQTLGen Consortium and GTEx Consortium studies). We observed Pearson's r=0.9 for cis-eQTL z-score overlap with eQTLGen, and Pearson's r=0.9 for trans-eQTLs. For cis-sQTLs, 89.0% of the 2,677 GTEX whole blood sGenes we also tested were |

also found as sGenes in our analysis. For trans-sQTLs, there were only 2 results in GTEX whole blood, both of which we replicated.

Randomization | Participants were randomly selected from the INTERVAL study, a cohort comprising presumably healthy blood donors, irrespective to covariate status.

Blinding | Blinding was not relevant to this study as there was no case-control status.

# Reporting for specific materials, systems and methods

We require information from authors about some types of materials, experimental systems and methods used in many studies. Here, indicate whether each material, system or method listed is relevant to your study. If you are not sure if a list item applies to your research, read the appropriate section before selecting a response.

## Materials & experimental systems

| n/a | Involved in the study |
|---|---|
| ☒ | ☐ Antibodies |
| ☒ | ☐ Eukaryotic cell lines |
| ☒ | ☐ Palaeontology and archaeology |
| ☒ | ☐ Animals and other organisms |
| ☒ | ☐ Clinical data |
| ☒ | ☐ Dual use research of concern |
| ☒ | ☐ Plants |

## Methods

| n/a | Involved in the study |
|---|---|
| ☒ | ☐ ChIP-seq |
| ☒ | ☐ Flow cytometry |
| ☒ | ☐ MRI-based neuroimaging |

## Plants

Seed stocks | N/A

Novel plant genotypes | N/A

Authentication | N/A

