## [Peer Review File · Nature Genetics]

The contribution of genetic determinants of blood gene expression and splicing to molecular phenotypes and health outcomes

Corresponding Author: Dr Dirk Paul

Version 0:

Decision Letter:

13th Mar 2024

Dear Dr Paul,

First, I am so sorry for the delay in returning this decision to you. Thank you so much for your patience, it is much appreciated!

Your Article, "Genetic determinants of blood gene expression and splicing and their contribution to molecular phenotypes and health outcomes" has now been seen by 3 referees. You will see from their comments below that while they find your work of interest, some important points are raised. We are interested in the possibility of publishing your study in Nature Genetics, but would like to consider your response to these concerns in the form of a revised manuscript before we make a final decision on publication.

We therefore invite you to revise your manuscript taking into account all reviewer comments. Please highlight all changes in the manuscript text file. At this stage we will need you to upload a copy of the manuscript in MS Word .docx or similar editable format.

*2) If you have not done so already please begin to revise your manuscript so that it conforms to our Article format instructions, available

[here](http://www.nature.com/ng/authors/article_types/index.html).

*3) Include a revised version of any required Reporting Summary: <https://www.nature.com/documents/hr-reporting-summary.pdf>

Please be aware of our [guidelines](https://www.nature.com/nature-research/editorial-policies/image-integrity) on digital image standards.

Link Redacted

We hope to receive your revised manuscript within four to eight weeks. If you cannot send it within this time, please let us know.

Sincerely,

Safia Danovi, PhD
Senior Editor, Nature Genetics
ORCID: 0009-0007-7822-5479

Referee expertise:

Referee #1: human genetics incl. gene regulation/expression

Referee #2: human genetics, multi-omic analyses

Referee #3: human genetics/genomics, computational biology

Reviewers' Comments:

Reviewer #1:

Remarks to the Author:

The manuscript by Tokolyi, Persyn et al describes the relationship between gene expression and splicing QTLs with protein, metabolomics, lipid and complex trait measurements. The most exciting feature of this manuscript is the large-scale data integration using RNA-sequencing from 4732 individuals and other large-scale omics with means to provide these data to the larger research community. The results on cis-eQTLs impacting trans-sQTL are also exciting. I have several comments and concerns:

- 1) The approaches to conduct trans-QTL analyses do not appear to appropriately account for mapping biases. See <https://www.ncbi.nlm.nih.gov/pmc/articles/PMC6305209/>
- 2) The approaches to conduct cis-eQTL effects would benefit from more comprehensively exploring cell-type-specific effects. See <https://pubmed.ncbi.nlm.nih.gov/27918533/>
- 3) Trans-QTL summary statistics should be made available. It would be much more preferable if they provide the subsetted tests and the matrix of all-by-all tests to support future replication activities of trans-e/sQTLs in the literature.
- 4) For the enrichment of cis-sQTLs near the TES, could this be due to 5'→3' bias in RNA-sequencing quantification?
- 5) Many of the colocalizations are from the same sets of individuals. Have the authors investigated whether this would inflate the number of colocalizations detected?
- 6) The authors should attempt replication of trans-eQTLs. They would have an impressive impact if they could test any of the cis-eSNPs that impact trans-splicing more directly (but I leave that as optional).
- 7) Enrichments of "nucleosome assembly" and "RNA polymerase II activity" are often generic GO terms. Have the authors more rigorously looked at splicing complex genes.
- 8) I found this statement confusing "with 2.03–95.55 million uniquely mapped reads (median: ~24 million)". Is this the number of reads/sample? It seems highly variable.
- 9) The methods description for the mediation analysis does not explicitly state they only did this in cis-QTLs so I had to infer that. They should state this explicitly, but this is just a minor clarification point.

Reviewer #2:

Remarks to the Author:

Overall a clear/well-written manuscript, and the portal and bulk download options are quite nice. Nice to see a larger eQTL/sQTL resource be made available. However, there are some gaps in the analysis that should be addressed. I disagree with the decision to focus on cis-QTLs only for the trans-QTL analysis- is it very likely that these are, in most cases, the only trans-QTLs we are currently powered to discover? Certainly possible. But I feel the authors should evaluate

this assumption/at least run some preliminary analyses- some trans-QTL signals are very strong. Also did any of your top PEER factors represent a trans eQTL hotspot (if so likely shouldn't be adjusted for in this analysis)?

The discussion of shared signals with other molecular traits was one of the most interesting parts of this work, as were the nice mediation analyses. Could the authors for the colocalized signals between proteomics and expression examine how this corresponds to reported cross-assay correlation coefficients, and shared QTLs, reported in <https://www.nature.com/articles/s41586-023-06563-x> and similar? This new expression data could add significantly to this debate, I'm sure you show more colocalizations than prior resources allowed for.

How did you decide on adjusting for 10 splicing PCs (for sQTLs) (I see Figure S6 for eQTLs)? Were any of your PEER factors mainly capturing cell type abundance/any issues with collinearity there? I also see in Methods some discussion of PC3 and PC4 (on expression data) but not the final PEER factors used in association analysis.

Why not examine a broader range of health outcomes from UK Biobank PheWAS or similar? This section felt quite limited with only 20 phenotypes from FinnGenn...

While as stated in Table S4, it is clear that cis eGenes are fairly maxed out in both eQTLGen/INTERVAL, a big strength of this projects design vs eQTLGen is ability to detect secondary tertiary etc signals, as well as build gene prediction models directly as opposed to in summary data. These issues should be explored more in the text. Not totally clear to me how this integrates with OmicsPred resource and final publication (if authors built prediction scores using all data in INTERVAL, if so where tested them, etc). It would also be helpful if Table S7 was at the individual signal, not just the gene, level. Finding distinct signals at same gene very important for colocalization work.

How does your work fit in with recent debates on whether eQTL are expected to be depleted for GWAS associations?

Could the authors add a section on replicating limited eQTLGen trans signals?

Minor: typo in Table S24- ="White blood cell (leukocyte) count (reported) (10"&charr(8313)&"/L)"

Reviewer #3:

Remarks to the Author:

Using the INTERVAL study dataset, Tokolyi and colleagues analyzed blood gene expression and splicing QTLs based on bulk short-read RNA-seq data of 4,732 individuals, and assessed the contribution of expression and splicing QTLs to molecular and phenotypic traits. Expression QTLs (eQTLs) and splicing QTLs (sQTLs) across human tissues including blood have been systematically studied before, for example by the Geuvadis and GTEx projects. Compared to previous studies, the major strengths of this work include the large sample size, and the availability of multiple other molecular measurements (e.g. proteomic and metabolomic traits) in the same cohort. Additionally, the authors analyzed multiple external datasets of molecular and phenotypic traits, including protein abundance data from deCODE and UKB-PPP, and health outcome data from FinnGen and COVID-19 HGI.

Major comments:

1. Despite a major focus on sQTLs, the presentation of sQTL results is vague and non-conventional. The authors used LeafCutter for splicing analysis, but the splicing results generated are not easy to interpret. For example, the authors stated in Line 134-138 that 128 splicing events were found in FAM157C, but only 9 of them overlap with previously defined exon boundaries. This number seems too low to be true and raises a concern about the validity of the LeafCutter/sQTL results. I suggest that the authors use a complementary event-based splicing tool (e.g. rMATS-turbo) to perform the sQTL analysis and compare results against LeafCutter. To present and visualize the splicing results, the authors should follow conventions in the field, including using the Percent Spliced In (PSI) metric and showing sashimiplot visualization of highlighted sQTL events. Additionally, throughout the manuscript, the authors seem to use "splicing events" to refer to "alternative splicing events". But these two phrases have very different meanings and should not be mixed.
2. Perhaps the most interesting biological finding in this work concerns the colocalization of sQTLs and protein QTLs (pQTLs). In particular, the authors honed into alternative splicing removal of transmembrane domains as a mechanism for how alternative splicing modulates the abundance of soluble proteins circulating in blood. Nonetheless, alternative splicing can modulate circulating protein levels through multiple alternative mechanisms, e.g. by affecting mRNA stability, mRNA translational efficiency, and protein stability. Can the authors investigate and comment on these alternative mechanisms for sQTL-pQTL colocalization?
3. Can the authors explicitly report how many sQTL-pQTL colocalization events involve alternative splicing removal of transmembrane domains? Of these, how many are in genes encoding single-pass vs multi-pass transmembrane proteins? Is there an enrichment compared to sQTLs that do not colocalize with pQTLs?
4. A major selling point of this study is that the large sample size increases the power for eQTL and sQTL discovery. To solidify this point and provide guidance for future studies of this kind, the authors should perform a down-sampling analysis to assess how eQTL/sQTL discovery is influenced by sample size.
5. The authors rightly pointed out that trans-eQTL/sQTL analysis requires very large sample size, thus the current analysis is under-powered. While this point is valid and well taken, more work is needed to confirm the events currently reported. For example, the authors reported that the splicing factor QKI is associated with 18 sGenes in trans. Can the authors investigate and confirm if QKI indeed regulates the splicing of these 18 genes? The authors may use publicly available RNA-seq data upon QKI perturbation (e.g. data from ENCODE). Alternatively, they can knockdown or knockout QKI in a blood cell line and assess the splicing of these 18 genes. Likewise, the authors should assess the effect of TFIP11 on ABHD3 splicing.

Additional specific comments:

Line 146-147: Please define what are “primary” and “secondary” cis-sQTLs.

Line 171-173: From a mechanistic perspective, I don't understand why it is relevant to investigate the distance to the TSS, for sQTLs colocalizing vs not colocalizing with eQTLs. Why not investigating whether the sQTLs introduce premature termination codon and potentially trigger nonsense-mediated decay, which is one of the mechanisms that may lead to sQTL-eQTL colocalization?

Line 181-196: It is understandable that cis-eGenes with eSNPs in trans-association could be enriched for TFs and C2H2 ZF-KRAB, but why would trans-eGenes be enriched for these categories too (Figure S4)? Please explain and clarify.

Line 227: “post-transcriptional molecular phenotypes” is a strange/inappropriate phrase for proteomic and metabolomic traits. Please revise.

Line 278-289: the numbers shown in the text do not seem to match the numbers shown in Figure 4B. For example, the number of Nightingale-measured metabolic traits mediated by cis-eQTLs seems to be too low in Figure 4B as compared to the text.

Line 301-307: can the authors confirm if this missense SNP in IL6R is also the lead SNP or in high LD with the lead SNP for the sQTL? I think this is what the authors suggested but the writing is not entirely clear. Please clarify.

Line 335: can the authors comment on why exon 10 splicing is associated with plasma protein level of WARS1?

Line 364-365: sashimiplot visualization of this sQTL event would be informative.

Line 394-395: the authors noted that “98 splicing events that colocalized with pQTLs (such as IL6R and IL7R) excised protein-coding sequences encoding transmembrane domains.” This number should have been mentioned in Results first.

Version 1:

Decision Letter:

7th Aug 2024

Dear Dr Paul,

Your Article, "Genetic determinants of blood gene expression and splicing and their contribution to molecular phenotypes and health outcomes" has now been seen by 3 referees. We are interested in the possibility of publishing your study in Nature Genetics, but would like to consider your response to their remaining requests in the form of a revised manuscript before we make a final decision on publication.

We therefore invite you to revise your manuscript taking into account all reviewer and editor comments. Please highlight all changes in the manuscript text file. At this stage we will need you to upload a copy of the manuscript in MS Word .docx or similar editable format.

*1) Include a “Response to referees” document detailing, point-by-point, how you addressed each referee comment. If no action was taken to address a point, you must provide a compelling argument. This response will be sent back to the referees along with the revised manuscript.

*2) If you have not done so already please begin to revise your manuscript so that it conforms to our Article format instructions, available

[here](http://www.nature.com/ng/authors/article_types/index.html).

*3) Include a revised version of any required Reporting Summary: <https://www.nature.com/documents/nr-reporting-summary.pdf>

Please be aware of our [guidelines](https://www.nature.com/nature-research/editorial-policies/image-integrity) on digital image standards.

Link Redacted

We hope to receive your revised manuscript within four to eight weeks. If you cannot send it within this time, please let us know.

Nature Genetics is committed to improving transparency in authorship. As part of our efforts in this direction, we are now requesting that all authors identified as 'corresponding author' on published papers create and link their Open Researcher and Contributor Identifier (ORCID) with their account on the Manuscript Tracking System (MTS), prior to acceptance. ORCID helps the scientific community achieve unambiguous attribution of all scholarly contributions. You can create and link your ORCID from the home page of the MTS by clicking on 'Modify my Springer Nature account'. For more information please visit please visit www.springernature.com/orcid.

Sincerely,

Safia Danovi, PhD
Senior Editor, Nature Genetics
ORCID: 0009-0007-7822-5479

Reviewers' Comments:

Reviewer #1:

Remarks to the Author:

The authors have sufficiently addressed my concerns.

Reviewer #2:

Remarks to the Author:

I am in general very pleased with the authors' response, particularly the revised "all by all" transQTL analysis. I think more of the response to review 2.2, including the figure and the identity of these 10 SNPs (this may help other eQTL efforts understand why some of the associations with these 10 pleiotropic loci are not identified in this study but may be in others, depending on ranking of PEER factors and how many PEER factors are adjusted for). Pretty minor, can definitely go in the supplement, but I think it should be explained to enhance comparability with future work.

Also in section 2.10. I think the authors response is very nice and thorough, but I don't think the citation to Mostafavi et al and the discussion made it in to the main text.

Reviewer #3:

Remarks to the Author:

The authors have responded to all of my comments and suggestions. There are two remaining issues that I'd like the authors to resolve.

1. A suggestion was made on the first submission to confirm the results from the trans-eQTL/sQTL analysis. Specifically, the authors were suggested to use publicly available RNA-seq data or perform perturbation experiments to investigate if splicing factors associated with sGenes in trans indeed regulate the splicing of these genes. In response to this suggestion the authors assessed QKI knockdown RNA-seq data from ENCODE. They found that 8 genes had alternative splicing differences upon QKI knockdown, although no data was shown in the revised manuscript or responses to reviewer comments. The authors also noted that TFIP11 was not in the ENCODE dataset. The authors concluded that "these results are hypothesis-generating and further experimental validation will be needed to confirm their mechanistic regulation", and that "additional experimental work will be required to confirm our presented results". However, I should clarify that what I suggested was to provide basic experimental validation of the trans-sQTL results. Specifically, in light of the results described in the responses to reviewer comments, the authors should go ahead to knockdown or knockout QKI and TFIP11 individually, perform RNA-seq to assess transcriptome changes, and match the results with their trans-sQTL results involving these two proteins. The work requested here is not complicated, but will go a long way in establishing the validity

of the results currently reported. The results from this analysis should be added to the revised manuscript.

2. The authors seem reluctant to display sashimiplot visualization of their sQTL results, for reasons that I don't agree with, as such a visualization has been routinely done in the splicing field including in past sQTL studies. Nonetheless, the authors did add a sashimiplot of IFNAR2 exons 7-9, stratified by genotype, but the plot is confusing. Can the authors clarify which exons/splicing events were called as differential among genotypes? From the sashimiplot provided, the exon density and splice junction coverage among genotypes seem largely comparable.

Version 2:

Decision Letter:

Our ref: NG-A64130R1

15th Oct 2024

Dear Dr Paul,

Thank you for submitting your revised manuscript "Genetic determinants of blood gene expression and splicing and their contribution to molecular phenotypes and health outcomes" (NG-A64130R1). It has now been seen by the original referees and their comments are below. The reviewers find that the paper has improved in revision, and therefore we'll be happy in principle to publish it in Nature Genetics, pending minor revisions to satisfy the referees' final requests and to comply with our editorial and formatting guidelines.

Sincerely,

Safia Danovi, PhD
Senior Editor, Nature Genetics
ORCID: 0009-0007-7822-5479

Reviewer #3 (Remarks to the Author):

The authors have addressed all of my remaining comments.

Genetic determinants of blood gene expression and splicing and their contribution to molecular phenotypes and health outcomes

Rebuttal letter

Reviewer #1:

The manuscript by Tokolyi, Persyn et al describes the relationship between gene expression and splicing QTLs with protein, metabolomics, lipid and complex trait measurements. The most exciting feature of this manuscript is the large-scale data integration using RNA-sequencing from 4732 individuals and other large-scale omics with means to provide these data to the larger research community. The results on *cis*-eQTLs impacting *trans*-sQTL are also exciting.

We thank the reviewer for the positive feedback on our manuscript.

I have several comments and concerns:

1.1. The approaches to conduct *trans*-QTL analyses do not appear to appropriately account for mapping biases (see <https://www.ncbi.nlm.nih.gov/pmc/articles/PMC6305209/>).

We thank the reviewer for pointing out the important issue described in the paper presented by Saha and Battle, 2018. In this article, the authors state: “Our results demonstrate that trans-eQTL associations in a standard pipeline are dominated by potential false-positives due to sequence similarity and replication rates between studies may be artificially inflated due to this pattern. Additionally, genes with sequence similarity display more correlated expression levels, and mapping errors should be considered in co-expression analysis as well. Our results do not imply that all instances of co-expression or trans-eQTL associations arising from genes with sequence similarity are in fact false positives. Genes with sequence similarity also sometimes have true functional relationships.”

Although we tried to minimize the likelihood of cross-mapping by only using uniquely mapping reads for gene expression quantification, we acknowledge that low levels of false positives due to cross-mapping may remain. To quantify the possible impact of cross-mapping on our trans-eQTL results, we considered whether our set of s/eGenes had elevated sequence similarity with other genes, as suggested by, and using the data provided in Saha and Battle, 2018. We computed symmetric cross-mappability resources for Gencode v33 in GRCh38, with exon k=75, UTR k=36, and up to 2 mismatches (Saha et al.). Of our reported trans-eQTLs, only 1,079 (12.21%) had >0 reported symmetric sequence similarity between the tested eGene and those regulated by the trans-eSNP in cis. Of the trans-sQTLs, 969 (23.56%) had >0 reported symmetric sequence similarity between genes. This is dramatically lower than the rates reported by Saha et al. (75%).

To acknowledge the potential impact of mapping biases on a subset of our findings, we have added a column to the Tables S9 and S11, summarizing the trans-s/eQTL results to indicate whether each trans-QTL has >0 sequence similarity. We have also commented on these results in the Methods section of the revised manuscript:

“The Bonferroni multiple-testing correction method (i.e., $p=0.05/\text{number of tested trans-associations}$) was applied to identify significant trans-associations. While previous work has demonstrated that *trans*-QTL analyses may be susceptible to artefacts due to read cross-mapping between similar genes (Saha and Battle, 2018), our quantification approach using only uniquely mapped reads led to only a small fraction of our *trans*-QTL results (i.e., 12.2% of *trans*-eQTLs and 23.6% of *trans*-sQTLs) involving genes that were flagged for sequence similarity.”

1.2. The approaches to conduct *cis*-eQTL effects would benefit from more comprehensively exploring cell-type-specific effects (see <https://pubmed.ncbi.nlm.nih.gov/27918533/>).

We thank the reviewer for raising this interesting question regarding the effect of cell proportions on our eQTL results. We recognize that gene expression in this bulk whole-blood RNA-seq cohort will be influenced by each individual’s cell composition. Therefore, in the original version of the manuscript, we have accounted for this variation through the inclusion of measured cell counts as covariates in our QTL models. In this way, the associations we present provide a comprehensive resource of average effects in blood that may be relevant to a wide range of diseases and disease states.

We agree with the reviewer that mapping cell-type-specific eQTLs is very informative, as exemplified by BLUEPRINT (Chen, et al., 2016). As such, we believe that carefully designed discovery studies with cell type specificity as the main analysis goal are preferred, as opposed to post-hoc cell type deconvolution from bulk data. Also, we assessed and integrated genetic effects across different molecular data layers (i.e., proteins, metabolites), all of which were generated in circulating blood (rather than within specific cell types). Further, with the emerging availability of population-scale single-cell data, we believe that this specific question will be more appropriately addressed in future analyses of other cohorts.

Nonetheless, we have highlighted the importance of this for downstream interpretation in the Discussion section of the revised manuscript:

“In our study, we further demonstrate the value of such a dataset when integrating data from multiple molecular phenotypes in the same individuals and linking these to external health outcomes to help address the variant-to-function challenge (e.g., through statistical colocalization and mediation analyses). Our analysis of bulk RNA-seq provides a comprehensive resource of average QTL effects across blood cell types. Similar application to population biobanks is warranted and with the emerging availability of concomitant molecular data at the single-cell level across a wide range of tissues, single-cell-QTL mapping at population scale will become feasible.”

1.3. *Trans*-QTL summary statistics should be made available. It would be much more preferable if they provide the subsetted tests and the matrix of all-by-all tests to support future replication activities of *trans*-e/sQTLs in the literature.

We agree with the reviewer that the full *trans*-QTL summary statistics could be useful for future replication analyses and have now added these to the web portal. We have edited the Results section of the revised manuscript to reflect this:

“Next, we investigated the distal (*trans*) regulatory effect of genetic variants, defined as >5Mb from the TSS/splicing event. First, we performed an untargeted, all-vs-all *trans*-eQTL and sQTL analysis. We found a high correlation of *trans*-eQTL Z-scores for SNP–gene pairs also tested by the eQTLGen Consortium study (Vosa, et al., 2021) (Pearson’s $r=0.9$, Figure S4). [...]

In our targeted analysis, we identified 2,058 *trans*-eGenes significantly associated with *cis*-eSNPs at the Bonferroni-corrected threshold of $p < 5 \times 10^{-11}$ (Figure 2B; Tables S1, S2 and S9).”

1.4. For the enrichment of *cis*-sQTLs near the TES, could this be due to 5’->3’ bias in RNA-sequencing quantification?

We appreciate the reviewer’s concern that the RNA library preparation method could have led to 3’ bias in the sequencing data. We investigated this using the ‘geneBody_coverage’ function from RSeQC (Wang, et al., 2012), which quantifies average coverage across housekeeper genes for each sample in the cohort. For the vast majority of samples, this showed that there was little bias in read mapping across the gene body (**Response Figure 1**).

Response Figure 1: Gene body coverage for each sample in the post-QC INTERVAL cohort, calculated using RSeQC.

1.5. Many of the colocalizations are from the same sets of individuals. Have the authors investigated whether this would inflate the number of colocalizations detected?

We thank the reviewer for pointing out this potential issue when performing colocalization analyses using datasets generated from the same sets of individuals. Before conducting the suggested analyses, we reviewed the literature and found evidence that, with a sufficiently large sample size, the impact of

using the same samples is negligible (Mitchelmore, et al., 2020). In the Methods section of our original manuscript, we referred to the paper in which the simulations were conducted (Figure S1 in Mitchelmore, et al., 2020). In the revised version, we have edited the Methods section to include the number of individuals used in the simulation to indicate that our cohort far exceeds the reported sample size of 200 individuals.

“A previous study performed simulations showing that the impact of complete sample overlap on colocalization results was negligible with 200 individuals and, therefore, will be even smaller with large sample sizes as used here.”

1.6. The authors should attempt replication of *trans*-eQTLs. They would have an impressive impact if they could test any of the *cis*-eSNPs that impact *trans*-splicing more directly (but I leave that as optional).

We agree with the reviewer that replication of the identified *trans*-QTLs in independent cohorts is important. We have now performed a replication analysis of the identified *trans*-eQTLs in INTERVAL with those identified by the eQTLGen Consortium (Phase 1). We tested *trans*-eQTLs present in both datasets. We found that the 7,481 overlapping SNP–gene pairs with $p \leq 1 \times 10^{-5}$ in INTERVAL and significant in eQTLGen have highly correlated Z-scores (Pearson’s $r=0.90$). We have added this observation to the Results section of the revised manuscript (see below), as well as a new Figure S4. Further details of this analysis are presented in **Response 2.11**.

We acknowledge that the presented replication analysis has its limitations (see **Response 2.11**). We highlight that we are currently involved in several collaborative efforts in which meta-analyses will increase confidence in *trans*-e/sQTL associations, i.e., eQTL Catalogue and eQTLGen Phase II. In particular, the forthcoming manuscript of eQTLGen Phase II comprises over 43,000 individuals across 52 datasets, including INTERVAL.

Finally, while we fully agree with the reviewer that experimental validation of the mechanisms underlying *cis*-eSNPs that impact *trans*-splicing would be insightful, we believe this to be beyond the scope of our paper.

We have commented further on this in the Discussion section of the revised manuscript:

“Our study has limitations. First, statistical power was limited to map genome-wide eQTLs and sQTLs in *trans*. As *trans*-QTLs are challenging to replicate and distinguish from cell type heterogeneity in bulk RNA-seq studies, we prioritized the identified conditionally independent lead *cis*-eSNPs for our *trans*-QTL analyses to prioritize the mechanism of upregulated gene expression modifying the expression and splicing of downstream genes. Whilst we show strong correlation of effect sizes for SNP–gene pairs also tested by the eQTLGen Consortium (Phase I) (Vosa, et al., 2021), large-scale meta-analyses of *trans*-QTL datasets will be required to create a resource of replicated associations, such as that being prepared by the eQTLGen Consortium (Phase II)

(<https://www.eqtngen.org/>). To aid in these efforts, we have provided full *trans*-QTL summary statistics on our web portal.”

1.7. Enrichments of “nucleosome assembly” and “RNA polymerase II activity” are often generic GO terms. Have the authors more rigorously looked at splicing complex genes.

We agree with the reviewer that these GO terms are relatively generic and do not provide specific mechanistic insight into our trans-sQTL associations. To address the reviewer’s concern, we have investigated an additional curated set of known splicing factors (Gerstberger, et al., 2014). We did not observe a significant enrichment for these factors in the cis-eGenes associated with the trans-sSNPs.

1.8. I found this statement confusing “with 2.03–95.55 million uniquely mapped reads (median: ~24 million).”. Is this the number of reads/sample? It seems highly variable.

*We thank the reviewer for highlighting this potential source of confusion. Due to the large cohort size, sequencing took place over a period of time and resulted in unequal pool/batch sizes, which influenced the number of reads obtained per sample. The originally reported range was the number of uniquely mapped reads assigned to gene features before any sample QC, resulting in the wide range of reported values. Most samples had comparable read depths (median=24.0 and IQR=21.5–26.9 million uniquely mapped reads, **Response Figure 2**). Two batches of samples were sequenced twice (to fill the sequencing flow-cell), and the reads pooled prior to mapping (range after removing batches 1 and 15 = 10.2–53.9 and IQR=21.3–26.2 million unique reads).*

In the revised manuscript, we have made the following edits to the Methods section to clarify the post-QC range of sequencing depth and added the IQR value for a better description of the distribution. Finally, we note that we have accounted for library size and batch in our analyses.

~~“**Gene expression quantification.** The raw gene level count data contained N=60,676 genes across N=4,778 individuals with 2.03–95.55 million uniquely mapped reads (median: ~24 million). Sequencing was performed across 15 batches.~~

Quality control of gene expression data. Sequencing was performed across 15 batches. We filtered out samples of poor quality by removing samples with a read depth below 10 million uniquely mapped reads. On average, each sample had 25.3 million unique reads (interquartile range, IQR=21.5–26.9, including batches 1 and 15 for which libraries were sequenced twice).”

Response Figure 2: Number of uniquely mapped reads assigned to gene features in each sample retained following QC, by sequencing batch.

1.9. The methods description for the mediation analysis does not explicitly state they only did this in *cis*-QTLs so I had to infer that. They should state this explicitly, but this is just a minor clarification point.

We have made the following edits to the revised Results and Methods sections to clarify that the mediation analyses were performed using the independent lead SNPs for *cis*-SNP–gene pairs only.

Results:

“To assess causality of the transcriptional phenotypes on downstream molecular phenotypes, we performed mediation analyses focusing on colocalizing molecular traits assayed in the INTERVAL study and SNP–gene/splicing event pairs identified from the *cis*-QTL analysis (Figure 4A; Methods).”

Methods:

“In the models, we defined (1) the independent lead *cis*-eQTL (or *cis*-sQTL) SNP (coded as 0, 1 and 2) as the independent (exposure) variable, (2) the gene expression level (or splicing event phenotype) of the *cis*-eGene/-sGene as the mediator and (3) the molecular trait as the dependent (outcome) variable.”

Reviewer #2:

Overall, a clear/well-written manuscript, and the portal and bulk download options are quite nice. Nice to see a larger eQTL/sQTL resource be made available.

We thank the reviewer for the positive feedback on our manuscript.

However, there are some gaps in the analysis that should be addressed.

2.1. I disagree with the decision to focus on *cis*-QTLs only for the *trans*-QTL analysis - is it very likely that these are, in most cases, the only *trans*-QTLs we are currently powered to discover? Certainly possible. But I feel the authors should evaluate this assumption/at least run some preliminary analyses - some *trans*-QTL signals are very strong.

We have now performed global trans-e/sQTL mapping to address this question. Using a global p-value threshold of $p < 5 \times 10^{-8} / 19,173$ (eQTL) and $p < 5 \times 10^{-8} / 111,937$ (sQTL) to reflect the total number of tests performed, we find 358,550 significantly associated SNP-gene pairs (for the expression of 2,312 genes), and 136,442 significantly associated SNP-splice event pairs (for 1,117 unique splice events in 509 sGenes).

*Comparing these results to our more targeted approach, we found that 1,897 genes and 613 splice events had significant associations in both analyses. Using LD blocks computed on GRCh38 [<https://doi.org/10.1101/2022.03.04.483057>] to infer the number of independent signals per gene, we observed that both the untargeted and targeted trans-eQTLs had significant associations in a median of 1 LD-block each. The same applied to trans-sQTLs. In total, there were 2,896 significant LD block-gene pairs for the targeted trans-eQTL analysis, and 3,621 significant for the untargeted analysis (802 and 1,602 for trans-sQTLs, respectively). This indicates that our focused analysis (A) captured most signals that we were powered to discover in this cohort; and (B) was also well designed to identify the strongest associations present. Further, the focused analysis enabled us to associate and prioritize potential upstream regulators of the trans-eQTLs and -sQTLs for hypothesis generation, particularly in the case of transcript splicing where these are less well known. However, we acknowledge that we may be missing additional associations with this approach. Therefore, we have provided our full summary statistics for all-vs-all trans-QTL as a resource (**Response 1.3**). As mentioned in **Response 1.6**, currently ongoing and future community efforts towards meta-analysis across cohorts will be critical for enabling well-powered trans-analyses.*

2.2. Also did any of your top PEER factors represent a *trans* eQTL hotspot (if so likely shouldn't be adjusted for in this analysis)?

We thank the reviewer for pointing this out, as it could indeed reduce the power to detect trans-eQTL associations. We tested the association between the 53,457 cis-eSNPs and the 35 PEER factors that we included as covariates. We found 10 SNPs significantly associated (on 5 chromosomes) with one of the PEER factors 7, 10, 12, 13 and 26. We found high correlations of Z-scores for trans-eQTL involving

these ten SNPs before and after removing these PEER factors from the model (Pearson correlation=0.8, **Response Figure 3**), and identified significant association with an additional 121 trans-eGenes.

We repeated our targeted trans-eQTL analysis (53,457 cis-eSNPs) after removing the five PEER factors from the list of covariates and found 8,318 trans-associations (1,951 trans-eGenes) instead of 8,834 (2,058 trans-eGenes) with an overlap of 7,406 (1,811 trans-eGenes). 41% of the newly identified trans-associations were involving the 10 SNPs associated with a PEER factor. After this comparison, we decided to leave these 5 PEER factors included in the model, as few SNPs were significantly associated with these PEER factors. Also, we could identify an overall higher number of trans-eGenes by integrating all PEER factors in the model.

For trans-sQTL analyses, we did not find any significant association between the cis-eSNPs and splice PCs.

Response Figure 3: Comparison of trans-eQTL Z-scores for ten SNPs which are significantly associated with a PEER factor, before and after removing five PEER factors associated with cis-eSNPs from model covariates. Pink lines are drawn at the significance threshold $p=5 \times 10^{-11}$.

2.3. The discussion of shared signals with other molecular traits was one of the most interesting parts of this work, as were the nice mediation analyses. Could the authors for the colocalized signals between proteomics and expression examine how this corresponds to reported cross-assay correlation coefficients, and shared QTLs, reported in <https://www.nature.com/articles/s41586-023-06563-x> and similar? This new expression data could add significantly to this debate, I'm sure you show more colocalizations than prior resources allowed for.

We thank the reviewer for the positive feedback on this analysis. We were very interested in the question of how our colocalization results involving Olink and SomaScan proteomic assays related to the cross-

assay correlations reported by Eldjarn, et al., 2023. Indeed, we observed significantly higher cross-array correlations for proteins with a colocalized e/sQTL-pQTL signal for both platforms, compared to those with a colocalized signal for only one platform (one-sided Wilcoxon rank-sum test; eQTL-pQTL pairs: $p=3.1\times 10^{-4}$; sQTL-pQTL pairs: $p=1.4\times 10^{-7}$).

We have modified the Results section in the revised manuscript to include our observations and added two Supplementary Tables for e/sQTL-pQTL pairs being analysed in both platforms.

“For Olink- and SomaScan-measured proteins, genetic effect directions were more consistent ($p=5.4\times 10^{-10}$, Fisher’s exact test) for colocalizing eQTL-pQTL pairs (78.9% with consistent effect directions) than non-colocalizing pairs (59.0%). The uncoupling of eQTLs and pQTLs has previously been observed and could be due, for example, to post-transcriptional or post-translational mechanisms.

Of the 99 eQTL-pQTL pairs (364 sQTL-pQTL pairs) analyzed for colocalization in both Olink and SomaScan platforms, we found that 45 (127) had a colocalized signal on both platforms; 19 (57) on the Olink platform only; and 9 (41) on the SomaScan platform only (Table S16-17). We annotated these colocalization results with cross-assay correlations reported by Eldjarn, et al., 2023 and found significantly higher cross-assay correlations for e/sQTL-pQTL pairs with a colocalized signal for both platforms, compared to eQTL-pQTL pairs with a colocalized signal for only one platform (eQTL-pQTL pairs: $p=3.1\times 10^{-4}$; sQTL-pQTL pairs: $p=1.4\times 10^{-7}$, one-sided Wilcoxon rank-sum test). This indicates that the differences we observed in colocalization results might be due to differences in protein measurements between the two platforms.

2.4. How did you decide on adjusting for 10 splicing PCs (for sQTLs) (I see Figure S6 for eQTLs)?

We used a similar approach to that illustrated in Figure S7 (former Figure S6) for the eQTL analysis to determine the optimal number of principal components to include in the sQTL mapping (New Figure S7B). We have added the following explanation to the revised Methods section:

“**PEER factor and splicing principal component analysis.** [...] For inclusion in the eQTL analysis, we selected the number of PEER factors based on two criteria: (i) discovery of the largest number of *cis*-eGenes and (ii) additional gain in *cis*-eGenes with incremental increase in PEER factors (Figure S7A). We found that the relationship between the increase in number of discovered *cis*-eGenes and incremental increase in PEER factors is similar to that observed in the GTEx whole-blood dataset. Therefore, we included 35 PEER factors in our eQTL analysis, consistent with GTEx. We used a similar approach to determine the optimal number of principal components to include in the sQTL analysis, testing 0–10 PCs. We found that *cis*-sQTL discovery only increased slightly with the number of PCs included with no obvious threshold (Figure S7B). Given that 10 PCs was established as a previous default for sQTL mapping (Li, et al., 2018), we opted to include 10 splicing principal components.”

Figure S7. Identification of the optimal number of PEER factors (latent variation estimated from gene expression data) and PCs to be included as covariates in the eQTL and sQTL analyses. A. PEER factors were chosen based on the maximum number of cis-eGenes discovered (Y-axis, left) and the gain in cis-eGenes with incremental increase in PEER factors (Y-axis, right). B. cis-sQTL discovery with inclusion of increasing numbers of principal components in the model.

2.5. Were any of your PEER factors mainly capturing cell type abundance/any issues with collinearity there? I also see in Methods some discussion of PC3 and PC4 (on expression data) but not the final PEER factors used in association analysis.

In the Methods section in the original version of the manuscript, we defined the use of PC3 and PC4 in the sample QC process. However, this was not related to the eQTL mapping. When calculating the PEER factors for the eQTL analysis, we included the quantified cell counts as covariates. Therefore,

these variables were taken into account during the learning step, meaning that the latent factors calculated capture orthogonal information to the cell counts and other covariates provided (age, sex, BMI).

To address this question, in the revised version of the manuscript, we have explored associations between PEER factors and genotype for our trans-eQTL analysis (see **Response 2.2**). We appreciate that the details of how PCs were used in sample QC could lead to confusion with the use of latent variables in the QTL analysis. Hence, we have condensed the information included in the Methods section as follows:

“The Match Bam to VCF (MBV) method from QTLTools was used to identify sample mix-ups and cross-contamination. MBV directly compares each aligned RNA-seq BAM file to all the genotypes in the VCF file and computes the proportion of concordant heterozygous and homozygous sites. To reduce computation time, we only focused on chromosome 1. Based on the concordance (close to 100%) between the genotype data and RNA-seq samples, we identified and corrected for 10 pairs of mislabeled samples. We removed 7 RNA-seq samples that did not show a clear high concordance (highest was <50%) with any particular genotype sample – either due to cross-contamination or the actual matching genotypes were not available. ~~In addition, principal component (PC) analysis was performed to determine whether the 10 pairs of samples were mislabeled in the technical covariate file. We linked PC3 with RIN values to confirm that RIN and other related technical covariates were recorded after the swap occurred. Hence, we corrected the sample IDs for the 10 pairs of mislabeled samples in both the count data and technical covariate file. We linked PC4 with sex information to confirm that sex and other biological covariates we recorded before the swap and did not require correction for mislabeling.~~”

2.6. Why not examine a broader range of health outcomes from UK Biobank PheWAS or similar? This section felt quite limited with only 20 phenotypes from FinnGen...

We thank the reviewer for suggesting that we broaden the range of health outcomes. We focused on the 20 most common phenotypes with relevance to the circulatory system in FinnGen, as the sample size of our molecular QTL dataset is insufficiently powered for a comprehensive PheWAS approach. In the original version of the manuscript, we conducted a preliminary assessment of a defined subset of health outcomes that resulted in a few biological vignettes that we discussed in more detail.

We have now also completed a PheWAS of all traits using the Open Target Genetics Portal (version 22.10) and uploaded the results to our web portal. We have updated the Results section to make the reader aware of these additional data:

“**Deconvoluting mechanisms of GWAS disease loci with transcriptional and molecular phenotypes.** [...] Finally, we also performed a genetic look-up analysis of all independent signals at the identified cis-eQTLs and the splicing event phenotypes with a cis-sQTL using data from the Open Target Genetics Portal (v22.10). We present the results on our web portal (<https://IntervalRNA.org.uk>).”

2.7. While as stated in Table S4, it is clear that *cis* eGenes are fairly maxed out in both eQTLGen/INTERVAL, a big strength of this projects design vs eQTLGen is ability to detect secondary tertiary etc signals, as well as build gene prediction models directly as opposed to in summary data. These issues should be explored more in the text.

We agree with the reviewer that the conditional analysis is a distinct strength of our study and thank the reviewer for suggesting that we expand upon this aspect of our results. We have made the following edits to the Discussion section to emphasize the utility of the well-powered conditional QTL analyses, which enabled optimal colocalization and mediation analyses with a number of other molecular traits and phenotypes.

*“To this end, we discovered eQTLs for 17,233 genes and sQTLs for 29,514 splicing phenotypes in 6,853 genes in peripheral blood through RNA-sequencing of 4,732 individuals. This included non-primary signals for 81% of *cis*-eGenes and 49% *cis*-sGenes, substantially increasing knowledge of the independent genetic determinants of gene expression in whole blood. Then, we combined these data with mQTL and pQTL data in the same participants of the INTERVAL study to map the genetic basis for disease phenotypes. Notably, 52% of independent eQTL colocalizations and 28% of significant mediations involved non-primary eQTL signals, and 31% of independent sQTL colocalizations and 30% of significant mediations involved non-primary sQTL signals, demonstrating the value of the conditional analysis performed.”*

*With regard to the building of gene prediction models, please see **Response 2.8**.*

2.8. Not totally clear to me how this integrates with OmicsPred resource and final publication (if authors built prediction scores using all data in INTERVAL, if so where tested them, etc).

We wish to clarify that the present manuscript and the OmicsPred publication (Xu, et al., 2023) are distinct and non-overlapping. Both papers use the multi-omics data derived from participants of the INTERVAL cohort but perform different analyses to address different research questions. For example, the relevant part of OmicsPred focuses on building genetic predictors of gene expression levels and performing a transcriptome-wide association study. We have referenced the OmicsPred portal in the Discussion as we have previously uploaded eQTL data from a subset of INTERVAL. To avoid confusion, we have edited this in the revised version of the Discussion to ensure it is clear for the reader that <https://IntervalRNA.org.uk> is the relevant portal for this work:

“All generated data are accessible to the scientific community via the INTERVAL QTL portal (<https://IntervalRNA.org.uk>) and the cognate OmicsPred portal⁴⁶ (<https://www.omicspred.org/>).”

To answer the reviewer’s question about the methodology employed in Xu et al., both internal and external validation were performed. In this paper, the last two batches of samples were held out for external validation purposes and the first four were used for eQTL mapping, genetic-score training and internal validation.

2.9. It would also be helpful if Table S7 was at the individual signal, not just the gene, level. Finding distinct signals at same gene very important for colocalization work.

We agree with the reviewer on the importance of resolving distinct signals for future signal colocalization. Table S7 comprises a summary on the gene level because the relevant analysis was limited to the gene level due to (1) the different variant sets tested in the two cohorts and (2) the summary statistics available for GTEx. In addition, the splice event clustering was performed separately within each cohort. Hence, different phenotypes were tested as splicing is called de-novo from the RNA-seq. The independent sQTL and eQTL loci called in our cohort are available in Tables S3 and S6, enabling colocalization analysis at the individual signal level (as described later in our Results section).

2.10. How does your work fit in with recent debates on whether eQTL are expected to be depleted for GWAS associations?

In a recent analysis, Mostafavi et al. assessed systematic differences in discovery of genetic effects on gene expression and complex traits (Mostafavi, et al., 2023). Colocalization of complex-trait-associated variants with molecular QTL signals has been considered an effective approach for identifying putative molecular mechanisms. However, only a small fraction of GWAS signals colocalize with previously reported eQTL signals. Mostafavi et al. found that this is due to the fact that “GWAS and eQTL mapping are powered to identify different types of variants.” Therefore, modified/broader QTL mapping approaches may be needed to increase discovery of the molecular mechanisms underlying GWAS associations.

We believe there are several strengths of our study that have led to increased detection of GWAS-molecular QTL colocalizations:

- *Mostafavi et al. state in their Discussion section “[...] larger sample sizes in eQTL mapping will help.” Our large gene expression dataset enabled the discovery of much smaller QTL effects (please see also **Response 3.4** for an exploration of power for cis-QTL across sample sizes).*
- *We have resolved a large number of non-primary signals (**Response 2.7**), which may represent more context-specific regulatory effects and, therefore, be more likely to colocalize with GWAS associations.*
- *Mostafavi et al. further note that “other types of molecular QTL assay, including chromatin QTLs and splicing QTLs, may help link additional variants to functional effects”. Our splicing QTL colocalization results demonstrate both overlapping and distinct signals captured by sQTLs vs eQTLs, as previously reported by Li, et al., 2018. This further increased the overlap between GWAS results and the variants that modify transcriptional phenotypes.*

Taken together, we argue that the yield of discovered eQTLs at GWAS loci can still be increased when taking into consideration ever increasing sample sizes, different cell types and states. In turn, this will improve colocalization between eQTL and GWAS data. Until then, we caution against conclusive statements on the depletion of eQTLs at GWAS loci.

Nonetheless, in the Discussion section of the revised manuscript, we have now also captured this important point and added a citation to the analysis presented by Mostafavi et al.:

“[...] Similar application to population biobanks is warranted and with the emerging availability of concomitant molecular data at the single-cell level across a wide range of tissues, single-cell-QTL mapping at population scale will become feasible. Such data will enable us to dissect gene regulatory networks at much greater resolution across specific cell types and dynamic processes. Together, these improved molecular QTL data will further enhance the interpretation of GWAS signals (Li, et al., 2018). Our genetic discoveries are publicly available in an open-access and interactive resource at <https://IntervalRNA.org.uk>.”

2.11. Could the authors add a section on replicating limited eQTLGen *trans* signals?

*We thank the reviewer for this suggestion. As implied by the reviewer, we note that this replication analysis is necessarily limited to the SNP–gene pairs included and tested for trans-eQTL in both datasets. Focusing on the significant trans-eQTL from eQTLGen (n=59,786) and using our all-vs-all trans-eQTL analysis (see **Response 1.3**) to maximise this overlap, we compared our results to those from eQTLGen Phase I on the basis of Z-scores. We found that the 7,481 overlapping SNP–gene pairs in INTERVAL ($p \leq 1 \times 10^{-5}$) and eQTLGen ($p \leq 8.3 \times 10^{-6}$) have highly correlated Z-scores (Pearson’s $r=0.90$). We note that the correlation of absolute Z-scores is lower (Pearson’s $r=0.57$).*

Next, we assessed the overlap of significant trans-associations in both studies. As the sample size is much lower in the INTERVAL compared to eQTLGen study, we have less statistical power. Of the 2,924 most significant trans-associations in eQTLGen ($p < 1 \times 10^{-20}$), we replicated 63% in INTERVAL at the significance threshold $p < 1 \times 10^{-6}$. We note that differences in data analysis could impact on differences in the detection of trans-associations. For example, in eQTLGen, a large proportion of the trans-associations were driven by blood cell composition effects (representing 64% of trans-eQTL SNPs, Vosa, et al., 2021). In contrast, in INTERVAL, we added 19 blood cell counts as covariates in the association model. We have added these observations to the Results section of the revised manuscript, as well as a new Figure S4 showing the replication analysis:

“As our study has lower statistical power than eQTLGen, we focused on replicating the 2,924 most significant *trans*-eQTL associations from eQTLGen with $p < 1 \times 10^{-20}$. Of these, we replicated 63% at $p < 1 \times 10^{-6}$ in INTERVAL. We note that the incomplete overlap may be due to differences in data analysis strategies, including accounting for blood cell counts in the association model (Vosa, et al., 2021).”

New Figure S4. Correlation of trans-eQTL Z-scores between INTERVAL and eQTLGen. Z-scores were plotted for SNP–gene pairs for which there is a significant trans association in eQTLGen and that were also available in our all vs all trans-eQTL results.

As described in **Response 1.6**, we have also edited our Discussion section to comment further on replication of trans-signals in eQTLGen.

2.12. Minor: typo in Table S24- ="White blood cell (leukocyte) count (reported) (10⁹&charr(8313)&"/L)"

We have corrected the formatting error in the revised version of the manuscript.

Reviewer #3:

Using the INTERVAL study dataset, Tokolyi and colleagues analyzed blood gene expression and splicing QTLs based on bulk short-read RNA-seq data of 4,732 individuals, and assessed the contribution of expression and splicing QTLs to molecular and phenotypic traits. Expression QTLs (eQTLs) and splicing QTLs (sQTLs) across human tissues including blood have been systematically studied before, for example by the Geuvadis and GTEx projects. Compared to previous studies, the major strengths of this work include the large sample size, and the availability of multiple other molecular measurements (e.g. proteomic and metabolomic traits) in the same cohort. Additionally, the authors analyzed multiple external datasets of molecular and phenotypic traits, including protein abundance data from deCODE and UKB-PPP, and health outcome data from FinnGen and COVID-19 HGI.

We thank the reviewer for highlighting the strengths of our study, including the large sample size, availability of multiple other molecular measurements in the same cohort, and integration of multiple external datasets.

3.1. Despite a major focus on sQTLs, the presentation of sQTL results is vague and non-conventional. The authors used LeafCutter for splicing analysis, but the splicing results generated are not easy to interpret. For example, the authors stated in Line 134-138 that 128 splicing events were found in FAM157C, but only 9 of them overlap with previously defined exon boundaries. This number seems too low to be true and raises a concern about the validity of the LeafCutter/sQTL results. I suggest that the authors use a complementary event-based splicing tool (e.g. rMATS-turbo) to perform the sQTL analysis and compare results against LeafCutter. To present and visualize the splicing results, the authors should follow conventions in the field, including using the Percent Spliced In (PSI) metric and showing sashimiplot visualization of highlighted sQTL events. Additionally, throughout the manuscript, the authors seem to use “splicing events” to refer to “alternative splicing events”. But these two phrases have very different meanings and should not be mixed.

We thank the reviewer for these comments and suggestions on our presentation of the sQTL results. We have addressed these in turn below.

The authors used LeafCutter for splicing analysis, but the splicing results generated are not easy to interpret. [...] I suggest that the authors use a complementary event-based splicing tool (e.g. rMATS-turbo) to perform the sQTL analysis and compare results against LeafCutter. For example, the authors stated in Line 134-138 that 128 splicing events were found in FAM157C, but only 9 of them overlap with previously defined exon boundaries. This number seems too low to be true and raises a concern about the validity of the LeafCutter/sQTL results.

In our original manuscript, FAM157C was highlighted specifically because it was an extreme example. For this gene, the annotation file used included 27 donor and 26 acceptor sites in total, indicating that this gene is expected to have many different excision events detected. The 128 excision events we detect involve 51 unique donor locations and 52 unique acceptor locations, of which 7 and 6 respectively were present in the reference gtf file (Gencode 33). While only 9 excision events match exactly to both an annotated donor and acceptor site, a further 20 match to either a known donor or known acceptor.

Across all 128 events, there were a median of 10 split reads per individual, with at least 34 individuals with ≥ 10 reads per event (median 2,961 individuals with ≥ 10 reads across all events), demonstrating that the novel events were well supported across the cohort. Additionally, when overlapping these 128 excision events with those observed in the GTEx whole blood ($n=670$) unfiltered junction counts, we observed that 127/128 events were detected in the unrestricted dataset.

Thus, as not to mislead the reader, we clarify that FAM157C is an extreme example and not globally representative of the splice phenotypes. Therefore, we have removed the details about the splicing of FAM157C in the Results section of the revised manuscript:

“This included 543 *cis*-sGenes that were not identified as *cis*-eGenes. ~~The long non-coding RNA FAM157C, which is involved in cell proliferation and induction of apoptosis,²¹ contained the most splicing events ($n=128$, within 11 clusters defined by shared splice donor or acceptor sites). While this gene is known to contain 33 exons, the splicing events were mostly intronic ($n=105/128$) and rarely overlapping previously defined exon boundaries ($n=9/128$).~~ Across all splicing events with *cis*-sQTLs, these had a median length of 1,549bp and excised a protein-coding sequence in 32.4% of cases (the remainder related to intronic and UTR excisions).”

Globally, a much higher proportion of splice phenotypes with an sQTL are annotated (i.e., 66.25% both donor and acceptor; 82.00% either/or), and 93.12% of these splice events are also observed in the GTEx whole blood unfiltered junction counts, reflective of our stringent filtering thresholds for splice phenotypes. Linking this to our downstream colocalizations, 16.61% of splice phenotypes that match both annotated donor and acceptor sites and 16.80% of those that match either an annotated donor or acceptor site colocalized with a molecular trait. However, those that contain both novel donor and acceptor regions colocalized 23.12% of the time, revealing that the inclusion of un-annotated events further assists in explaining the shared architecture with downstream molecular and disease phenotype. We hope this reassures the reviewer with respect to the validity of LeafCutter, a method referenced in hundreds of studies, and our sQTL results. LeafCutter also enables the modelling of more complex splicing events than certain other tools, which we demonstrate as a benefit in this context of overlap with various downstream molecular and disease phenotypes. Nonetheless, we have added the following limitation to the Discussion section of the revised manuscript:

“Our study has limitations. First, while we have focused on one definition of transcript splicing due to the annotation-free approach benefiting the downstream analyses, other methodologies may shed light on other aspects of transcript splicing. Second, statistical power was limited to map genome-wide eQTLs and sQTLs in *trans*. [...]”

To present and visualize the splicing results, the authors should follow conventions in the field, including using the Percent Spliced In (PSI) metric and showing sashimplot visualization of highlighted sQTL events.

We thank the reviewer for this comment and would clarify that we are following the approach of previous high-profile sQTL studies (GTEx Consortium, 2020): our PSI values were inverse-normal transformed prior to sQTL analysis and, hence, results are quantified in this manner to relate to the

*beta-values provided for each sQTL. Additionally, sQTLs were computed separately for each splicing excision event and colocalized in turn, which is why figures explore each splice event independently as opposed to using sashimi plots. Additionally, due to our large sample size, sashimi plots are extremely crowded with many splice events observed in small numbers of individuals, detracting from the main messages of each example being the shared genetic architecture per splice event. Therefore, we have not produced such plots for every sQTL result. However, as suggested by the reviewer, we have added a sashimi plot to illustrate a specific example (see **Response 3.13**).*

Additionally, throughout the manuscript, the authors seem to use “splicing events” to refer to “alternative splicing events”. But these two phrases have very different meanings and should not be mixed.

Following the conventions defined in Li, et al., 2018, we define splice events in the Introduction as “de novo transcript splicing phenotypes (herein referred to as “splicing events”) from differential intron usage ratios” as assessed from split reads (junction counts) using the LeafCutter pipeline, excluding constitutively expressed junctions as standard.

I suggest that the authors use a complementary event-based splicing tool (e.g. rMATS-turbo) to perform the sQTL analysis and compare results against LeafCutter.

While we appreciate the reviewer’s suggestion to run rMATS-turbo, we do hope that our above clarification (that FAM157C was selected as an extreme example and the global LeafCutter results were much more in line with annotated junctions) will reassure the reader about the validity of LeafCutter. While it is possible in bioinformatics to run many different methods for the same research question, we believe the scale and depth of our INTERVAL data gives us the best opportunity to assess novel alternative splicing events and therefore an analysis using LeafCutter and presenting a single set of results maximizes both clarity and impact.

3.2. Perhaps the most interesting biological finding in this work concerns the colocalization of sQTLs and protein QTLs (pQTLs). In particular, the authors honed into alternative splicing removal of transmembrane domains as a mechanism for how alternative splicing modulates the abundance of soluble proteins circulating in blood. Nonetheless, alternative splicing can modulate circulating protein levels through multiple alternative mechanisms, e.g. by affecting mRNA stability, mRNA translational efficiency, and protein stability. Can the authors investigate and comment on these alternative mechanisms for sQTL-pQTL colocalization?

We thank the reviewer for the interest in this aspect of the manuscript. We note that as our dataset is derived from short-read sequences, we are limited in the conclusions we can draw about the consequence of the subsequent full isoforms the splice events will be constituents of. Additionally, the proteomic data was measured in plasma, so membrane proteins were of particular interest. We have edited our Discussion section to acknowledge these mechanisms as followed:

“Second, our analyses comprised proteins quantified in plasma, rather than intracellular proteins. Thus, the interpretation of the effects of gene expression and splicing QTLs on proteins may be due to impacts on both quantity and solubility of the resulting protein, and other regulatory mechanisms, such as the stability of the mRNA and protein in addition to translational efficiency, may not be captured. However, additional data would be needed to address this.”

3.3. Can the authors explicitly report how many sQTL-pQTL colocalization events involve alternative splicing removal of transmembrane domains? Of these, how many are in genes encoding single-pass vs multi-pass transmembrane proteins? Is there an enrichment compared to sQTLs that do not colocalize with pQTLs?

We found 166 independent sQTL signals involving alternative splicing removal of a transmembrane domain colocalized with a pQTL. Of these, 7 colocalization events involved multi-pass rather than single-pass transmembrane proteins. The colocalizing sQTL were not enriched for transmembrane domain removal compared to sQTL that did not colocalize with pQTL. We have edited the Results section of the revised manuscript to address this suggestion as follows:

“Further, we observed a pQTL colocalizing with an sQTL for the excision of a transmembrane domain in the encoding mRNA in 69 proteins, with 60.2% of these independent sQTL signals (n=100/166) not colocalizing with eQTLs for the same gene (Table S16). For example, this is observed in alpha-1 antitrypsin encoded by *SERPINA1*, and apolipoprotein L1 encoded by *APOLI1*. Of these 69 transmembrane proteins, the majority were annotated as being single-pass with only four (*ENTPD1*, *ADGRE2*, *ADGRE5*, *ADGRE1*) being multi-pass transmembrane proteins.”

3.4. A major selling point of this study is that the large sample size increases the power for eQTL and sQTL discovery. To solidify this point and provide guidance for future studies of this kind, the authors should perform a down-sampling analysis to assess how eQTL/sQTL discovery is influenced by sample size.

We thank the reviewer for this suggestion and have performed the down-sampling analysis to assess how sample size impacts cis-eQTL and cis-sQTL discovery. The results are now shown in New Figure S8 (and reproduced below). We find that these results agree with a previous simulation study by Huang, et al., 2018, which includes some of the authors of the current manuscript. The Huang et al study assessed power for eQTL detection using simulated datasets up to a sample size of 5,000 individuals and indicates that our complete INTERVAL dataset is well-powered to detect weak associations for common variants. This is consistent with the down-sampling results where our cis-eQTL discovery begins plateauing at >2,500 individuals. As expected, the cis-sQTL discovery will plateau at greater sample sizes, thus providing even greater support for large-scale studies of splicing. We have revised the Methods and Discussion sections as indicated below:

Methods:

“For each significant gene (or splicing event), a nominal p-value threshold was estimated to identify significant SNPs. To demonstrate how increased sample size assists in *cis*-eQTL and -sQTL discovery, random samples of patients in N=500 increments were subsetted and QTL mapping performed with the same inputs as the full *cis*-QTL analyses, with the output being the number of significant genes (or splice phenotypes) with a significant eQTL (sQTL). [...]”

Discussion:

“Then, we combined these data with mQTL and pQTL data in the same participants of the INTERVAL study to map the genetic basis for disease phenotypes. [...] Finally, we performed a down-sampling analysis to provide guidance towards the expected eQTL and sQTL discovery for future studies (Figures S8A and S8B).”

New Figure S8. *cis*-eQTL and *cis*-sQTL discovery rate as a function of sample size. Down-sampling analysis was performed for decreasing subsets of individuals from the full cohort. The percentage of genes with a significant *cis*-eQTL (A) and splice events with a *cis*-sQTL (B) was subsequently calculated.

3.5. The authors rightly pointed out that *trans*-eQTL/sQTL analysis requires very large sample size, thus the current analysis is under-powered. While this point is valid and well taken, more work is needed to confirm the events currently reported. For example, the authors reported that the splicing factor QKI is associated with 18 sGenes in *trans*. Can the authors investigate and confirm if QKI indeed regulates the splicing of these 18 genes? The authors may use publicly available RNA-seq data upon QKI perturbation (e.g. data from ENCODE). Alternatively, they can knockdown or knockout QKI in a blood cell line and assess the splicing of these 18 genes. Likewise, the authors should assess the effect of TFIP11 on ABHD3 splicing.

We appreciate the reviewer’s suggestions for targeted follow-up of the *trans*-sQTL networks. We accessed QKI knockdown RNA-seq data in K562 and HepG2 cells (Van Nostrand, et al., 2020) and

found that 12/18 *trans*-sGenes associated with QKI had been previously demonstrated to be targets of QKI. Of these, 8 genes had measured alternative splicing differences upon QKI knockdown. While this is reassuring in this specific example, we acknowledge these results are hypothesis-generating and further experimental validation will be needed to confirm their mechanistic regulation. Further, we note that TFIP11 was not tested by Van Nostrand, et al., 2020, as it represents a spliceosome component rather than a transcription factor. Additional experimental work will be required to confirm our presented results.

3.6. Line 146-147: Please define what are “primary” and “secondary” *cis*-sQTLs.

We have edited the text in our Results section to define these terms explicitly:

“After conditional analysis for each *cis*-sQTL, we identified 47,050 independent signals (34,205 unique lead variants), with a median of one independent signal per *cis*-sQTL (range: 1–20; Tables S1 and S6). To characterize independent variant effects on transcript splicing, we compared primary (i.e., the most significant independent QTL) with secondary (i.e., all other independent QTLs) *cis*-sQTLs.”

3.7. Line 171-173: From a mechanistic perspective, I don’t understand why it is relevant to investigate the distance to the TSS, for sQTLs colocalizing vs not colocalizing with eQTLs. Why not investigating whether the sQTLs introduce premature termination codon and potentially trigger nonsense-mediated decay, which is one of the mechanisms that may lead to sQTL-eQTL colocalization?

We were interested in any differences between the sQTLs that colocalized with an eQTL and those that did not, to begin investigating the potential mechanisms underlying eQTL-sQTL colocalization. We hypothesized that sQTLs that colocalized with eQTLs were likely to be regulated co-transcriptionally and, hence, may be located in regions more proximal to the TSS of the gene.

As noted in **Response 3.2**, we were limited by the nature of short-read data and, therefore, were not able to comprehensively assess effects that relate to the full isoform context such as NMD.

3.8. Line 181-196: It is understandable that *cis*-eGenes with eSNPs in *trans*-association could be enriched for TFs and C2H2 ZF-KRAB, but why would *trans*-eGenes be enriched for these categories too (Figure S4)? Please explain and clarify.

In reference to Figure S5 (former Figure S4), we assessed the enrichment for TFs and C2H2 ZF-KRAB amongst *trans*-eGenes for completeness. Given the similarity of these results to those for *cis*-eGenes, we believe our observations would be of interest to the reader; however, we would agree that the mechanism is unclear. We have added in the Results section that the *trans*-eGene results warrant further investigation.

“We investigated protein domain annotations for the observed transcription factors and detected a significant enrichment for the C2H2 zinc finger domain ($p=9.74\times 10^{-9}$ after Bonferroni multiple testing correction), specifically with the Krüppel-associated box (KRAB) domain ($p=3.04\times 10^{-10}$; Figure S5). For example, the *PLAG1* gene, which is an important regulator of the human hematopoietic stem cell dormancy and self-renewal, codes for a protein with a C2H2 zinc finger domain. We also noted the same enrichment for *trans*-eGenes ($p=5.73\times 10^{-5}$), although the molecular mechanisms are unclear and need further investigation.”

3.9. Line 227: “post-transcriptional molecular phenotypes” is a strange/inappropriate phrase for proteomic and metabolomic traits. Please revise.

In the revised version of the manuscript, we have used “proteomic and metabolomic traits” instead, as suggested by the reviewer.

3.10. Line 278-289: the numbers shown in the text do not seem to match the numbers shown in Figure 4B. For example, the number of Nightingale-measured metabolic traits mediated by *cis*-eQTLs seems to be too low in Figure 4B as compared to the text.

We thank the reviewer for bringing this to our attention, the figure displayed in the panel was that of an older version and this has now been corrected.

Updated Figure 4B: Total number of detected molecular phenotypes mediated by sQTLs and eQTLs.

“The expression of 143 *cis*-eGenes significantly mediated the effect of 413 *cis*-eSNPs on 202 downstream molecular phenotypes, including 101 SomaScan-measured proteins, 54 Olink-measured proteins, 39 Nightingale-measured metabolites, and 8 Metabolon-measured metabolites. In total, this comprised 525 significant eQTL mediation models (variant-gene-molecular phenotype triplets)

(Figure 4B). Similarly, we observed 106 splicing event phenotypes in 47 sGenes that significantly mediated the effect of 152 *cis*-sSNPs on 50 downstream molecular phenotypes, including 32 SomaScan-measured proteins, 16 Olink-measured proteins, 1 Nightingale-measured and 1 Metabolon-measured metabolite, comprising 241 significant sQTL mediation models.”

3.11. Line 301-307: can the authors confirm if this missense SNP in IL6R is also the lead SNP or in high LD with the lead SNP for the sQTL? I think this is what the authors suggested but the writing is not entirely clear. Please clarify.

This is correct, and we have clarified the Results section of the revised manuscript accordingly.

“In line with this finding, we observed the previously mentioned IL6R transmembrane splicing event mediating a minority of the effect of the lead SNP (rs12126142), which is in high LD ($r^2 > 0.99$; $D' > 0.99$) with this missense variant on Olink-measured plasma protein abundance (4.67%, $p = 1.12 \times 10^{-4}$) (Figures 4C and 4D).”

3.12. Line 335: can the authors comment on why exon 10 splicing is associated with plasma protein level of WARS1?

Whilst primarily found in the cytoplasm, secreted tryptophanyl-tRNA synthetase (WARS1) plays a role in the innate immune response. Nguyen, et al., 2023 have described two secretion pathways for WARS1: (1) Trp-dependent secretion of the naked protein, and (2) Trp-independent release from plasma-membrane-derived vesicles (PMVs). In the first pathway, Trp binding induces a conformational change that permits PI(4,5)P2 binding and subsequent release of WARS1 from the plasma membrane. In the second, CAPN2 is recruited and activated by calcium, leading to the formation of PMVs. Increased retention of exon 10 may, therefore, lead to more plasma WARS1 through an effect on binding of a factor involved in either secretion pathway. Exon 10 encodes amino acid residues 372-418, which does not include the Trp-binding domain nor any specific residues tested by Nguyen et al. for an effect on PI(4,5)P2 binding. CAPN2 was found to be bound by the C-terminal 154-471 amino acids of WARS1 but the role of the residues encoded by exon 10 were not explored. Therefore, we cannot confidently explain the precise mechanism through which exon 10 impacts secretion without additional experiments, which we feel would be beyond the scope of this paper.

3.13. Line 364-365: sashimiplot visualization of this sQTL event would be informative.

*As there are multiple splice events within IFNAR2 colocalizing with the COVID19-HGI GWAS traits, each with their own independent signals, this result is difficult to capture succinctly in a single sashimi plot (see also **Response 3.1**). To show an example figure in response to the reviewer’s suggestion, we have generated a sashimi plot (New Figure S9) for a subset of these sQTLs surrounding IFNAR2 exon 9. While 10 splice phenotypes across IFNAR2 colocalized with the COVID-HGI summary statistics, Figure S9 represents the cluster of events around exons 7-9, stratified by the genotypes of the lead SNP*

for the colocalizing signal with the highest PP H4. In order to properly reflect the data underlying our sQTL results, we have plotted the normalized residualized splice ratios used as the phenotype for the tensorQTL analysis. We have added this figure as an exemplar to the Supplementary Information of the revised manuscript.

New Figure S9. Sashimi plot demonstrating the cluster of splice events surrounding exons 7-9 of IFNAR2 with the highest posterior probability of colocalization with the COVID19-HGI GWAS, annotated with median splice event phenotype value per genotype.

3.14. Line 394-395: the authors noted that “98 splicing events that colocalized with pQTLs (such as IL6R and IL7R) excised protein-coding sequences encoding transmembrane domains.” This number should have been mentioned in Results first.

We thank the reviewer for pointing this out, and have added the below text to the revised Results section:

“Further, we observed a pQTL colocalizing with an sQTL for the excision of a transmembrane domain in the encoding mRNA in 69 proteins (98 unique splicing events), [...]”

We have edited the Discussion section as follows:

“For example, the 98 splicing events that colocalized with pQTLs (such as IL6R and IL7R) excised protein-coding sequences encoding transmembrane domains.”

References

1. Saha, A. & Battle, A. False positives in trans-eQTL and co-expression analyses arising from RNA-sequencing alignment errors. *F1000Res* **7**, 1860 (2018).
2. Chen, L. *et al.* Genetic Drivers of Epigenetic and Transcriptional Variation in Human Immune Cells. *Cell* **167**, 1398-1414 e24 (2016).
3. Vosa, U. *et al.* Large-scale cis- and trans-eQTL analyses identify thousands of genetic loci and polygenic scores that regulate blood gene expression. *Nat Genet* **53**, 1300-1310 (2021).
4. Wang, L., Wang, S. & Li, W. RSeQC: quality control of RNA-seq experiments. *Bioinformatics* **28**, 2184-5 (2012).
5. Mitchelmore, J., Grinberg, N.F., Wallace, C. & Spivakov, M. Functional effects of variation in transcription factor binding highlight long-range gene regulation by epromoters. *Nucleic Acids Res* **48**, 2866-2879 (2020).
6. Gerstberger, S., Hafner, M. & Tuschl, T. A census of human RNA-binding proteins. *Nat Rev Genet* **15**, 829-45 (2014).
7. Eldjarn, G.H. *et al.* Large-scale plasma proteomics comparisons through genetics and disease associations. *Nature* **622**, 348-358 (2023).
8. Li, Y.I. *et al.* Annotation-free quantification of RNA splicing using LeafCutter. *Nat Genet* **50**, 151-158 (2018).
9. Xu, Y. *et al.* An atlas of genetic scores to predict multi-omic traits. *Nature* **616**, 123-131 (2023).
10. Mostafavi, H., Spence, J.P., Naqvi, S. & Pritchard, J.K. Systematic differences in discovery of genetic effects on gene expression and complex traits. *Nat Genet* **55**, 1866-1875 (2023).
11. GTEx Consortium. The GTEx Consortium atlas of genetic regulatory effects across human tissues. *Science* **369**, 1318-1330 (2020).
12. Huang, Q.Q., Ritchie, S.C., Brozynska, M. & Inouye, M. Power, false discovery rate and Winner's Curse in eQTL studies. *Nucleic Acids Res* **46**, e133 (2018).
13. Van Nostrand, E.L. *et al.* A large-scale binding and functional map of human RNA-binding proteins. *Nature* **583**, 711-719 (2020).
14. Nguyen, T.T.T. *et al.* Tryptophan-dependent and -independent secretions of tryptophanyl- tRNA synthetase mediate innate inflammatory responses. *Cell Rep* **42**, 111905 (2023).

Genetic determinants of blood gene expression and splicing and their contribution to molecular phenotypes and health outcomes

Rebuttal letter

Reviewer #1:

The authors have sufficiently addressed my concerns.

We thank the reviewer for the helpful comments, which have substantially improved the manuscript.

Reviewer #2:

2.1. I am in general very pleased with the authors' response, particularly the revised "all by all" trans-QTL analysis.

We thank the reviewer for suggesting the all-by-all analysis, which will provide a helpful resource to the community.

2.2. I think more of the response to review 2.2, including the figure and the identity of these 10 SNPs (this may help other eQTL efforts understand why some of the associations with these 10 pleiotropic loci are not identified in this study but may be in others, depending on ranking of PEER factors and how many PEER factors are adjusted for). Pretty minor, can definitely go in the supplement, but I think it should be explained to enhance comparability with future work.

*As suggested by the reviewer, we have added Response Figure 3 from the previous Rebuttal Letter to the Supplementary Information of the revised manuscript (New **Figure S10**). We have also added the identity of the 10 SNPs significantly associated with any of the PEER factors to the legend of Figure S10.*

New Figure S10. Comparison of trans-eQTL Z-scores for ten SNPs significantly associated with a PEER factor, before and after removing the five PEER factors associated with cis-eSNPs from model covariates. Ten cis-eSNPs were found significantly associated with 5 PEER factors (Bonferroni multiple testing correction: $p < 0.05 / 53,457 \text{ cis-eSNPs} / 35 \text{ PEER factors} = 2.7 \times 10^{-8}$): rs7310615, rs17630235, rs3809272 (chromosome 12) associated with PEER factor 10; rs74505413, rs144476978, rs4795085, rs75715226 (chromosome 17) associated with PEER factor 12; rs11168070 (chromosome 5) associated with PEER factor 26; rs72878029 (chromosome 11) associated with PEER factor 13; and rs9264670 (chromosome 6) associated with PEER factor 7. The Pearson correlation of Z-scores for trans-eQTL involving these ten SNPs before and after removing these PEER factors from the model is 0.80. Pink lines are drawn at the significance threshold of $p=5 \times 10^{-11}$ for the identification of trans-eQTLs.

Further, we have revised the Methods section as follows (p20):

“For the *trans*-QTL analyses, we also assessed if there were PEER factors or splicing PCs associated with *cis*-eSNPs. We did not detect any significant associations ($p < 9.4 \times 10^{-8}$, Bonferroni multiple-testing correction across 53,457 *cis*-eSNPs and 10 splicing PCs) between *cis*-eSNPs and splicing PCs. However, we found that 10 *cis*-eSNPs were significantly associated with 5 PEER factors ($p < 2.7 \times 10^{-8}$, Bonferroni multiple-testing correction across 53,457 *cis*-eSNPs and 35 PEER factors). As a sensitivity analysis, we performed the *trans*-eQTL analyses with and without these 5 PEER factors. We showed a high correlation of Z-scores for *trans*-eQTL involving these ten SNPs before and after removing these PEER factors from the model (Pearson correlation = 0.80; Figure S10). We identified significant associations of *cis*-eSNPs with an additional 121 *trans*-eGenes. Overall, we identified a higher number of *trans*-eGenes by integrating all 35 PEER factors in the model (i.e., 2,058 *trans*-eGenes instead of 1,811 *trans*-eGenes).”

2.3. Also, in section 2.10. I think the authors response is very nice and thorough, but I don't think the citation to Mostafavi et al and the discussion made it into the main text.

We thank the reviewer for the positive feedback on our response. We have added the citation to Mostafavi et al and the discussion from the Rebuttal Letter to the Discussion section of the revised manuscript (p15):

“Together, these improved molecular QTL data will further enhance the interpretation of GWAS signals.¹ While GWAS signals have previously been observed to be depleted for eQTLs,² we demonstrate that the broader approaches employed in this study, such as the increased sample size, the resolution of non-primary signals, and the additional signals captured by the sQTLs, have the potential to increase discovery of the molecular mechanisms underlying GWAS association.”

Reviewer #3:

3.1. The authors have responded to all of my comments and suggestions.

We thank the reviewer for the constructive comments on our manuscript.

There are two remaining issues that I'd like the authors to resolve.

3.2. A suggestion was made on the first submission to confirm the results from the trans-eQTL/sQTL analysis. Specifically, the authors were suggested to use publicly available RNA-seq data or perform perturbation experiments to investigate if splicing factors associated with sGenes in trans indeed regulate the splicing of these genes. In response to this suggestion the authors assessed QKI knockdown RNA-seq data from ENCODE. They found that 8 genes had alternative splicing differences upon QKI knockdown, although no data was shown in the revised manuscript or responses to reviewer comments. The authors also noted that TFIP11 was not in the ENCODE dataset. The authors concluded that “these results are hypothesis-generating and further experimental validation will be needed to confirm their mechanistic regulation”, and that “additional experimental work will be required to confirm our presented results”. However, I should clarify that what I suggested was to provide basic experimental validation of the trans-sQTL results. Specifically, in light of the results described in the responses to reviewer comments, the authors should go ahead to knockdown or knockout QKI and TFIP11 individually, perform RNA-seq to assess transcriptome changes, and match the results with their trans-sQTL results involving these two proteins. The work requested here is not complicated, but will go a long way in establishing the validity of the results currently reported. The results from this analysis should be added to the revised manuscript.

We agree with the reviewer that the association results we have presented would require further investigation to confirm the regulatory mechanisms involved. The experimental work required for that would ideally be part of a future paper focused on locus-specific gene regulatory logic.

The reviewer's original request was either to “use publicly available RNA-seq data upon QKI perturbation (e.g. data from ENCODE)” or “knockdown or knockout QKI in a blood cell line and assess the splicing of these 18 genes.” Further, the reviewer suggested that we “assess the effect of TFIP11 on

ABHD3 splicing". In our original response to this request, we appreciate that we did not address the reviewer's request of providing experimental validation of the trans-sQTL results for QKI (transcription factor) and TFIP11 (spliceosome component). However, instead, we analysed the relevant dataset prepared by Van Nostrand et al., as suggested by the reviewer.

In response to the reviewer's new comment, we have now added the data to the Results section of the manuscript and expanded our systematic comparison to the Van Nostrand et al. data (New Table S12). As QKI was the only gene observed as a trans-sQTL regulator in our analysis, as well as tested for knockdown consequences by ENCORE,³ we limited the replication to this gene. Specifically, for all trans-sGenes for the QKI cis-eSNP we assessed the presence of a significant eCLIP binding peak in the ENCORE data (and the cell-type in which this was observed), as well as significant differential splicing upon QKI knockdown. Additionally, we observed that eCLIP peaks within ENCORE that were trans-sQTL genes in our analysis had a significantly lower p-value ($p=0.021$, one-sided Wilcoxon test) than those that were not.

This observation was also consistent when only assessing the ENCORE K562 data ($p=0.012$). Likewise, within the ENCORE alternative splicing data, we observed that genes that were trans-sQTL genes in our analysis had a significantly greater absolute Δ PSI ($p=0.021$) and significantly lower p-values ($p=0.002$) upon QKI knockdown than those that were not. A significant enrichment was not observed in the K562-only alternative splicing data, though we caution this had the least overlap ($n=3$ genes).

Taken together, while for a single trans-sQTL, these data provide confidence that our data and conclusions are robust.

We have added these new results to the revised manuscript (p7) and prepared New Table S12:

"For ABHD3, we demonstrate in addition that this trans-sSNP is also a cis-eSNP for the splicing factor TFIP11 and its antisense lncRNA TFIP11-DT, potentially regulating the splicing of this gene in trans. To provide experimental validation, we overlapped our trans-sQTL associations with the RBP binding and alternative splicing upon knockdown analyses in K562 and HepG2 cells performed by the ENCORE project.³ As QKI was the only gene also tested by the authors, we limited our analysis to this gene (Table S12). We found that 12/18 sGenes were reported as eCLIP targets of QKI, and 8/18 of these reported significant alternative splicing differences in ENCORE. Further, we observed that genes that were trans-sQTL sGenes in our analysis had more significant eCLIP binding ($p=0.021$, one-sided Wilcoxon test) and a significantly greater absolute Δ PSI upon knockdown ($p=0.021$) compared to those that were not. Across the whole dataset, cis-eGenes of trans-sSNPs were significantly enriched for 10 GO terms, including "nucleosome assembly" ($p=2.78 \times 10^{-6}$) and "RNA polymerase II activity ($p=1.40 \times 10^{-5}$) (Table S10)."

We are mindful that experimental validation of individual, targeted findings of our large dataset may not yield a meaningful outcome, as one should be cautious to generalise to the wider set of discoveries. In addition, our QTLs data were derived from whole blood, a substantially heterogeneous tissue, making their validation in cell lines that usually represent individual cell types challenging (despite accounting

for differences in cell proportions in our discovery analyses). We note limitations in the availability of appropriate experimental models for the validation of whole-blood QTLs.

INTERVAL trans-sQTL genes for QKI	QKI binding targets (eCLIP peaks)	ENCORE alternative splicing changes upon QKI knockdown
FYB1 / FYB	K562	K562
PICALM	HepG2, K562	HepG2, K562
NIN	HepG2	
BAZ2B	HepG2	
ELOVL5	HepG2, K562	
PACSIN2	HepG2	HepG2
NUMB	HepG2, K562	HepG2
MARK2	HepG2	HepG2
STXBP5	HepG2, K562	
TPM3	HepG2	
ERBIN / ERBB2IP	HepG2, K562	
ABI1	HepG2	K562
ANKRD44		
BNIP2		HepG2
AC092755.1 / BNIP2		HepG2
DYSF		
MACROH2A1		
CCNDBP1		

New Table S12. Overlap of sGenes regulated in trans by QKI cis-eSNPs with eCLIP peaks (representing RBP targets) and genes alternatively spliced upon knockdown in ENCORE.

3.3. The authors seem reluctant to display sashimiplot visualization of their sQTL results, for reasons that I don't agree with, as such a visualization has been routinely done in the splicing field including in past sQTL studies. Nonetheless, the authors did add a sashimiplot of IFNAR2 exons 7-9, stratified by genotype, but the plot is confusing. Can the authors clarify which exons/splicing events were called as differential among genotypes? From the sashimiplot provided, the exon density and splice junction coverage among genotypes seem largely comparable.

To ensure that readers can visualise relevant plots (e.g. sashimi) for any sQTL they may be interested in, we have made the residualised splicing phenotypes and raw junction ratios available on the web portal and via EGA, respectively.

We concur that the *IFNAR2* result is difficult to interpret as it is, due to the complex nature of the associations at this locus, involving multiple independent signals colocalising with several different COVID-19 HGI phenotypes. To clarify further, the sashimi plot shows the result that, as the dosage of the alternative allele (A) increases, we observe increased excision of the region surrounding *IFNAR2* exon 9 (highlighted in **Revised Figure S6**), and decreased excision of surrounding introns. We have added annotations to the plot to clarify how the genotype (rs2284550) is associated with differential splicing across this locus (**Revised Figure S6**). We note that this represents only one of several transcript splicing phenotypes and independent signals that colocalize at this locus; yet, it is a useful illustration of the complexity of the data for the reader and hopefully enables further deep-dive analyses.

We also agree that the coverage of the different splice phenotypes appears quite similar by eye in this plot. Given the size of our dataset, we can detect small but highly significant changes in alternative splicing associated with common variants. In the plotted case, the median normalized excision ratio increases from -0.07 (G/G) to 0.28 (A/A) for the transcript splicing phenotype excising part of the exon and decreases from 0.19 (G/G) to -0.08 (A/A) for the competing intron. As demonstrated by the colocalisation and mediation results throughout the manuscript, such associations can still be disease-relevant and, therefore, are of particular interest. To better illustrate the sQTL association, we have added inserts to this figure panel showing paired boxplots of the genotype-splice phenotype associations, adjusted for all other covariates included in the sQTL model, as well as the COVID-19 HGI GWAS result at this location.

Revised Figure S6. Transcriptional mechanisms underlying the *IFNAR2* locus associated with COVID-19 susceptibility and severity. A. Sashimi plot demonstrating the cluster of splice events surrounding exons 7-9 of *IFNAR2*. These represent those with highest posterior probability of colocalization with the COVID19-HGI GWAS, annotated with median splice event phenotype value per genotype. This is grouped by the minor allele count for the lead SNP of the splice event with the highest posterior probability of colocalization. B. Boxplot for the lead SNP for the transcript splicing phenotype excising part of the last exon. C. QTL plot of the variants at this signal for both the transcript splicing phenotype and the COVID-HGI GWAS.

References

1. Li, Y.I. *et al.* Annotation-free quantification of RNA splicing using LeafCutter. *Nat Genet* **50**, 151-158 (2018).
2. Mostafavi, H., Spence, J.P., Naqvi, S. & Pritchard, J.K. Systematic differences in discovery of genetic effects on gene expression and complex traits. *Nat Genet* **55**, 1866-1875 (2023).
3. Van Nostrand, E.L. *et al.* A large-scale binding and functional map of human RNA-binding proteins. *Nature* **583**, 711-719 (2020).